# SCOT: IMPROVED TEMPORAL COUNTERFACTUAL ESTIMATION WITH SELF-SUPERVISED LEARNING

## ABSTRACT

Estimation of temporal counterfactual outcomes from observed history is crucial for decision-making in many domains such as healthcare and e-commerce, particularly when randomized controlled trials (RCTs) suffer from high cost or impracticality. For real-world datasets, modeling time-dependent confounders is challenging due to complex dynamics, long-range dependencies and both past treatments and covariates affecting the future outcomes. In this paper, we introduce Self-supervised Counterfactual Transformer (SCOT), a novel approach that integrates self-supervised learning for improved historical representations. The proposed framework combines temporal and feature-wise attention with a component-wise contrastive loss tailored for temporal treatment outcome observations, yielding superior performance in estimation accuracy and generalization to out-of-distribution data compared to existing models, as validated by empirical results on both synthetic and real-world datasets.

## 1 INTRODUCTION

Accurate estimation of treatment outcomes over time conditioning on the observed history is a fundamental problem in causal analysis and decision making in various applications (Mahar et al., 2021; Ye et al., 2023; Wang et al., 2023). For example, in medical domains, doctors are interested in knowing how a patient reacts to a treatment or multi-step treatments; in e-commerce, retailers are concerned about how future sales change if adjusting the price of an item. While randomized controlled trials (RCTs) are the gold standard for treatment outcome estimation, most often than not such trials are either too costly or even impractical to conduct. Therefore, utilizing available observed data (such as electronic health records (EHRs) and historical sales) for accurate treatment outcome estimation, has drawn increasing interest in the community.

Compared to the well-studied i.i.d cases, treatment outcome estimation from time series observations not only finds more applications in the real world but also pose significant more challenges, due to the complex dynamics and the long-range dependencies in time series. Existing works along this endeavors explore various architectures with improved capacity and training strategies to alleviate time-dependent confounding[1]. Recurrent marginal structural networks (RMSNs) (Lim, 2018), counterfactual recurrent networks (CRN) (Bica et al., 2020), and G-Net (Li et al., 2021) utilize architectures based on recurrent neural networks. To mitigate time-dependent confounding, they train proposed models with inverse probability of treatment weighting (IPTW), treatment invariant representation through gradient reversal, and G-computation respectively, in addition to the factual estimation loss on observed data. Causal Transformer (CT) (Melnychuk et al., 2022) further improves capturing long-range dependencies in the observational data with a tailored transformer-based architecture and overcomes the temporal confounding with balanced representations trained through counterfactual domain confusion loss.

While existing methods achieve performance gain in empirical evaluation, they rely on the fully supervised loss of future outcomes to learn representations of history and thus suffer from its limitations. In many practical applications, we are confronted with the cold case challenge, where no or limited observations of testing time series are accessible. For example, after training the model with historical sales (seen as outcomes) and pricing (seen as treatments) sequences of products in

---

[1]We leave a more detailed review of related work in Sec. A of the appendix.

the food category, we are asked to estimate the sales of new items in the household category given their pricing, with no or very limited observations of household items collected beforehand. Existing methods based on supervised learning have difficulty generalizing to different domains and handling cold cases in test time.

In this work, we propose a paradigm shift from supervised learning to self-supervised training for temporal treatment outcome estimation. Our proposed model, **Self-supervised Counterfactual Transformer (SCOT)**, addresses the aforementioned limitations. To enhance the model capacity, we propose an encoder architecture composed of alternating temporal and feature-wise attention, capturing dependencies among both time steps and features. To learn expressive and transferable representations of the observed history, we refine the contrastive loss in self-supervised learning to a finer-grained level: both the entire history and each of the covariate/treatment/outcome components are contrasted when constructing the loss. Moreover, we view the counterfactual outcome estimation problem from the unsupervised domain adaptation (UDA) perspective and provide the theoretical analysis of the error bound for a counterfactual outcome estimator that gives estimation based on representations from self-supervised learning.

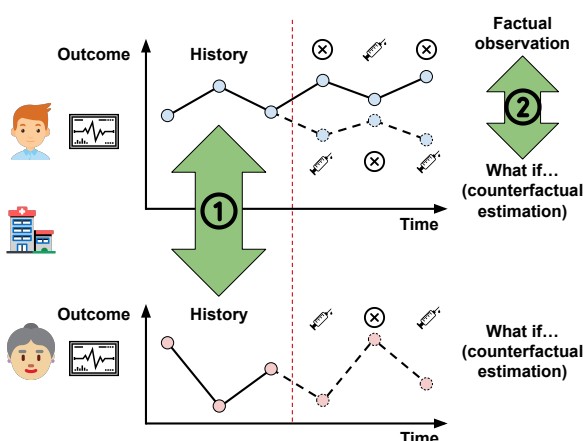

Figure 1: We illustrate the problem of treatment outcome estimation over time with an example in healthcare. The symbol cross/needle refers to the value of a binary treatment (giving the medicine/doing nothing). Dashed lines refer to the results of counterfactual outcome estimation. We propose SCOT as a temporal counterfactual estimator enhanced with self-supervised learning, inducing transferrability to both ① cold-start cases from unseen subpopulations and ② counterfactual outcome estimation.

Our main contributions are summarized as follows:

1. We adapt self-supervised learning (SSL) together with component-wise contrastive losses tailored for temporal observations to learn more expressive representations of the history in temporal counterfactual outcome estimation.

2. We propose a new encoder architecture combining both temporal attention and feature-wise attention for better modeling the complex temporal dependencies and feature interactions in observations.

3. Empirical results show that our proposed framework outperforms existing baselines across both synthetic and real-world datasets in both estimation accuracy and generalization. In addition, we demonstrate that the learned representations are balanced towards treatments and thus address the temporal confounding issue.

## 2 PROBLEM FORMULATION

Our task is estimating the outcomes of subjects with observed history after being applied a sequence of treatments from observational data (Lim, 2018; Bica et al., 2020; Melnychuk et al., 2022). We represent the available observed dataset as $\{\{\boldsymbol{x}_t^{(i)}, \boldsymbol{a}_t^{(i)}, \boldsymbol{y}_t^{(i)}\}_{t=1}^{T^{(i)}}, \boldsymbol{v}^{(i)}\}_{i=1}^{N}$ of $N$ independently sampled subjects, where $T^{(i)} \in \mathbb{N}^{+}$ denotes the length of the observed history of subject $i$, and $\boldsymbol{x}_t^{(i)} \in \mathbb{R}^{d_X}$, $\boldsymbol{a}_t^{(i)} \in \mathbb{R}^{d_A}$, and $\boldsymbol{y}_t^{(i)} \in \mathbb{R}^{d_Y}$ stand for the observed vector of covariates, treatments, and outcomes respectively, at time $t$ of subject $i$. $\boldsymbol{v}^{(i)} \in \mathbb{R}^{d_V}$ contain all static features of subject $i$. We omit the subject index $i$ in the following text for notational simplicity.

Following the potential outcomes (Splawa-Neyman et al., 1990; Rubin, 1978) framework extended to time-varying treatments and outcomes (Robins & Hernan, 2008), our target is to es-

timate $\mathbb{E}(\boldsymbol{y}_{t+\tau}[\bar{\boldsymbol{a}}_{t:t+\tau-1}]|\bar{\boldsymbol{H}}_t)$ for $\tau \geq 1$, where $\bar{\boldsymbol{H}}_t = (\bar{\boldsymbol{X}}_t, \bar{\boldsymbol{A}}_{t-1}, \bar{\boldsymbol{Y}}_t, \boldsymbol{V})$ is the observed history. $\bar{\boldsymbol{X}}_t = (\boldsymbol{x}_1, \boldsymbol{x}_2, \ldots, \boldsymbol{x}_t)$, $\bar{\boldsymbol{A}}_{t-1} = (\boldsymbol{a}_1, \boldsymbol{a}_2, \ldots, \boldsymbol{a}_{t-1})$, $\bar{\boldsymbol{Y}}_t = (\boldsymbol{y}_1, \boldsymbol{y}_2, \ldots, \boldsymbol{y}_t)$, $\boldsymbol{V} = \boldsymbol{v}$. $\bar{\boldsymbol{a}}_{t:t+\tau-1} = (\boldsymbol{a}_t, \boldsymbol{a}_{t+1}, \ldots, \boldsymbol{a}_{t+\tau-1})$ is the sequence of the applied treatments in the future $\tau$ discrete time steps. In factual data, $\bar{\boldsymbol{H}}_t$ and $\boldsymbol{a}_t$ are correlated, leading to the treatment bias in counterfactual outcome estimation. In addition, the distribution of $\bar{\boldsymbol{H}}_t$ can also vary between training and test data: $P_{\mathcal{D}_{tr}}(h) \neq P_{\mathcal{D}}(h)$, causing the feature distribution shifts. Following the tradition in domain adaptation, we name $P_{\mathcal{D}_{tr}}(h)$ and $P_{\mathcal{D}}(h)$ the source/target domains respectively. Table 4 describes feature distribution shifts in our datasets.

To ensure the identifiability of treatment effects from observational data, we take the standard assumptions used in existing works (Bica et al., 2020; Melnychuk et al., 2022): (1) consistency, (2) positivity and (3) sequential strong ignorability (See Appendix B).

## 3 SELF-SUPERVISED COUNTERFACTUAL TRANSFORMER

We illustrate the detailed design of our proposed Self-supervised Counterfactual Transformer (SCOT). Our main goal is to learn representations of observed history sequences that are informative for counterfactual treatment outcome estimation, which we achieve by tailoring both the representation encoder architecture and the self-supervised training loss. On top of the representation learning, we also propose a simple yet effective decoder for non-autoregressive outcome prediction and demonstrate improvements in both the accuracy and the speed of multi-step estimation. Fig. 2 overviews the proposed framework.

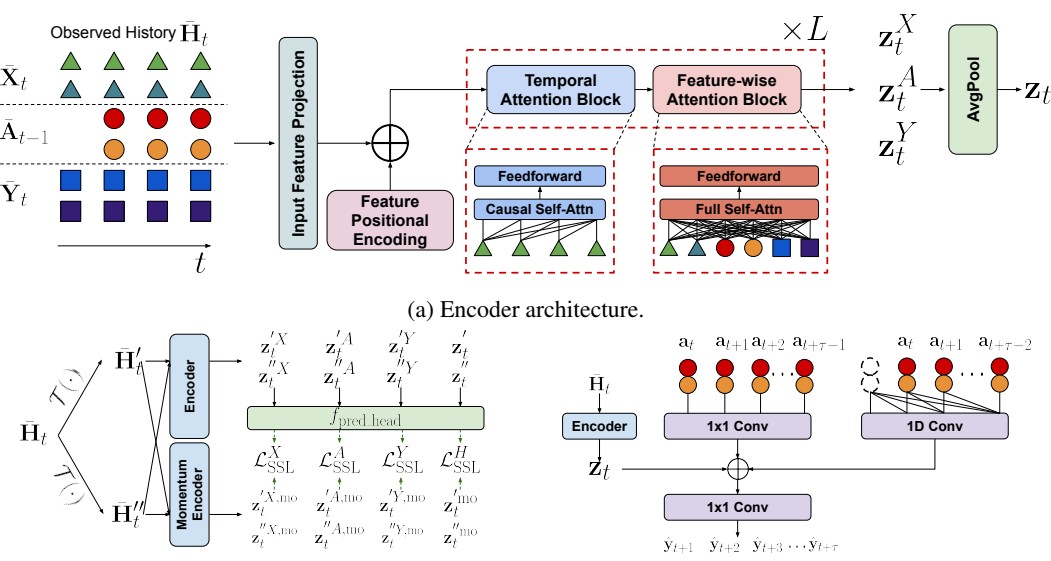

(a) Encoder architecture.

(b) Self-supervised learning of history.

(c) Non-autoregressive outcome predictor.

Figure 2: Overview of SCOT. (a) Encoder architecture. The Temporal Attention Block applies temporal causal attention along the time dimension in parallel for each feature, while the Feature-wise Attention Block calculates full self-attention along the feature dimension in all time steps. (b) Self-supervised learning of the history representations. Positive pairs are generated by applying random transformations $\mathcal{T}(\cdot)$ on the same sample. We construct component-wise contrastive losses of historical covariates, treatments and outcomes in addition to the standard contrastive loss of the entire sequence. (c) Non-autoregressive outcome predictor architecture.

### 3.1 ENCODER ARCHITECTURE

For a given sequence of the observed history $\bar{\boldsymbol{H}}_t \in \mathbb{R}^{t \times d_{\text{input}}}$ concatenated from $(\bar{\boldsymbol{X}}_t, \bar{\boldsymbol{A}}_{t-1}, \bar{\boldsymbol{Y}}_t)$ (we omit static variables $\boldsymbol{V}$ here for simplicity and leave the processing details in Sec. 3.1; $d_{\text{input}} = d_X + d_A + d_Y$), the encoder in Fig. 2a maps the entire history to representations $\{\boldsymbol{z}_t^i \in \mathbb{R}^{d_{\text{model}}}\}_{i=1}^{d_{\text{input}}}$ for each feature $f_i$. Then, we employ average pooling for feature-wise representations from corresponding

features to get the representations of covariate, treatment, and outcome components, and all features for the representation of the entire observed history. Denote the set of covariate, treatment, and outcome variables as $\mathbf{F}_X$, $\mathbf{F}_A$, and $\mathbf{F}_Y$ respectively. We have:

$$\boldsymbol{z}_t^X = \mathrm{avg}(\{\boldsymbol{z}_t^i\}_{f_i \in \mathbf{F}_X}),\ \boldsymbol{z}_t^A = \mathrm{avg}(\{\boldsymbol{z}_t^i\}_{f_i \in \mathbf{F}_A}),\ \boldsymbol{z}_t^Y = \mathrm{avg}(\{\boldsymbol{z}_t^i\}_{f_i \in \mathbf{F}_Y}),\ \boldsymbol{z}_t = \mathrm{avg}(\{\boldsymbol{z}_t^i\}_{i=1}^{d_{\mathrm{input}}}).\tag{1}$$

**Input feature projection.** Assume the concatenation of time-varying variables $(\bar{\boldsymbol{X}}_t, \bar{\boldsymbol{A}}_{t-1}, \bar{\boldsymbol{Y}}_t)$ in history as $\bar{\boldsymbol{S}}_t \in \mathbb{R}^{t \times d_S}$, $d_S = d_X + d_A + d_Y$, and the static variables $\boldsymbol{V} \in \mathbb{R}^{d_V}$. We adopt a linear transformation $f_{\mathrm{input}} : \mathbb{R} \to \mathbb{R}^{d_{\mathrm{model}}}$ to map $\bar{\boldsymbol{S}}_t$ and $\boldsymbol{V}$ to the embedding space as $\boldsymbol{E}^S \in \mathbb{R}^{t \times d_S \times d_{\mathrm{model}}}$ and $\boldsymbol{E}^V \in \mathbb{R}^{d_V \times d_{\mathrm{model}}}$ respectively, where

$$\boldsymbol{E}^S[i,j] = f_{\mathrm{input}}(\bar{\boldsymbol{S}}_t[i,j]),\ 1 \le i \le t,\ 1 \le j \le d_S;\tag{2}$$

$$\boldsymbol{E}^V[j] = f_{\mathrm{input}}(\boldsymbol{V}[j]),\ 1 \le j \le d_V.\tag{3}$$

**Feature positional encoding.** A shared feature projection function among all features is not sufficient to encode the feature-specific information since the same scalar value represents different semantics in different features. Meanwhile, feature-specific information is also critical for modeling the interactions among features. Therefore we enhance the input embedding with a positional encoding along the feature dimension. Since the features can be grouped into covariates, treatments and outcomes and form a hierarchical structure with 2 levels, we model it with a learnable tree positional encoding (Shiv & Quirk, 2019; Wang et al., 2021). Denote the lists of covariate, treatment, outcome, and static features as $\boldsymbol{F}_X$, $\boldsymbol{F}_A$, $\boldsymbol{F}_Y$, and $\boldsymbol{F}_V$. For the $i$-th feature $f_i^{\boldsymbol{F}}$ in a certain feature list $\boldsymbol{F} \in \{\boldsymbol{F}_X, \boldsymbol{F}_A, \boldsymbol{F}_Y, \boldsymbol{F}_V\}$, its positional encoding is:

$$\boldsymbol{E}_{\mathrm{fea\_pos}}(f_i^{\boldsymbol{F}}) = \boldsymbol{E}_{\mathrm{fea}} \cdot \mathrm{Concat}(\boldsymbol{e}_{\boldsymbol{F}}, \boldsymbol{e}_i),\ \text{where}\ \boldsymbol{e}_{\boldsymbol{F}} = \left\{ \begin{array}{l} (1,0,0,0)\ \text{if}\ \boldsymbol{F}\ \text{is}\ \boldsymbol{F}_X \\ (0,1,0,0)\ \text{if}\ \boldsymbol{F}\ \text{is}\ \boldsymbol{F}_A \\ (0,0,1,0)\ \text{if}\ \boldsymbol{F}\ \text{is}\ \boldsymbol{F}_Y \\ (0,0,0,1)\ \text{if}\ \boldsymbol{F}\ \text{is}\ \boldsymbol{F}_V \end{array} \right.,\tag{4}$$

$\boldsymbol{e}_i \in \mathbb{R}^{\max(d_X, d_A, d_Y, d_V)}$ is a one-hot vector with only $\boldsymbol{e}_i[i] = 1$. $\boldsymbol{E}_{\mathrm{fea}} \in \mathbb{R}^{d_{\mathrm{model}} \times (4 + \max(d_X, d_A, d_Y, d_V))}$ are learnable tree embedding weights. After obtaining the stacked feature positional embeddings $\boldsymbol{E}_{\mathrm{fea\_pos}}^S \in \mathbb{R}^{d_S \times d_{\mathrm{model}}}$, $\boldsymbol{E}_{\mathrm{fea\_pos}}^V \in \mathbb{R}^{d_V \times d_{\mathrm{model}}}$ of time-varying and static features, we broadcast them to the shape of $\boldsymbol{E}^S$ and $\boldsymbol{E}^V$ respectively along the time dimension. The embedded input is then the sum of input feature projection and feature positional encoding:

$$\boldsymbol{Z}^{S,(0)} = \boldsymbol{E}^S + \mathrm{Broadcast}(\boldsymbol{E}_{\mathrm{fea\_pos}}^S),\quad \boldsymbol{Z}^{V,(0)} = \boldsymbol{E}^V + \mathrm{Broadcast}(\boldsymbol{E}_{\mathrm{fea\_pos}}^V).\tag{5}$$

**Temporal attention block.** The temporal attention block is designed to capture the temporal dependencies within each feature. We construct the block based on the self-attention part in the conventional Transformer decoder (Vaswani et al., 2017). Considering the importance of relative time interval in modeling treatment outcomes, we adopt the relative positional encoding (Shaw et al., 2018; Melnychuk et al., 2022) along the time dimension.

The temporal attention block in the $l$-th layer receives $\boldsymbol{Z}^{S,(l-1)}$ from the previous layer, reshapes it to $d_S$ sequences with lengths $t$, and passes them through the block in parallel. The outputs $\boldsymbol{Z}_{\mathrm{tmp}}^{S,(l)}$ have the same shape as $\boldsymbol{Z}^{S,(l-1)}$. Since $\boldsymbol{Z}^{V,(l-1)}$ is static, we only pass it through the point-wise feed-forward module and get $\boldsymbol{Z}_{\mathrm{tmp}}^{V,(l)}$

**Feature-wise attention block.** The feature-wise attention block models interactions among different features. We reuse the architecture of the conventional Transformer encoder but replace the positional encoding with the feature positional encoding as described. The block in the $l$-th layer receives $\boldsymbol{Z}_{\mathrm{tmp}}^{S,(l)}$ from the temporal attention block and reshapes it to $t$ sequences, each with a length $d_S$. We broadcast $\boldsymbol{Z}_{\mathrm{tmp}}^{V,(l)}$ and concatenate it with each sequence along the feature dimension to enable the attention among both time-varying and static features. The concatenated $t$ sequences that we apply attention to are:

$$\boldsymbol{Z}_{\mathrm{tmp}}^{SV,(l)} = \mathrm{Concat}(\boldsymbol{Z}_{\mathrm{tmp}}^{S,(l)}, \mathrm{Broadcast}(\boldsymbol{Z}_{\mathrm{tmp}}^{V,(l)})) \in \mathbb{R}^{t \times (d_S + d_V) \times d_{\mathrm{model}}}.\tag{6}$$

We apply full attention across all features and get $\boldsymbol{Z}^{SV,(l)}$ with the same shape as $\boldsymbol{Z}^{SV,(l)}_{\text{tmp}}$. The propagated embeddings of time-varying features are obtained as:

$$\boldsymbol{Z}^{S,(l)} = \boldsymbol{Z}^{SV,(l)}[:, :d_S, :] \in \mathbb{R}^{t \times d_S \times d_{\text{model}}}. \tag{7}$$

To keep the $\boldsymbol{Z}^{V,(l)}$ static after feature-wise attention, we only propagate $\boldsymbol{Z}^{V,(l)}_{\text{tmp}}$ with full attention among static features only. The updated embeddings of the static features $\boldsymbol{Z}^{V,(l)} \in \mathbb{R}^{d_V \times d_{\text{model}}}$ have the same shape as $\boldsymbol{Z}^{V,(l)}_{\text{tmp}}$.

## 3.2 SELF-SUPERVISED REPRESENTATION LEARNING OF THE OBSERVED HISTORY

We employ pretraining for the encoder in a self-supervised way with the contrastive learning objectives $\mathcal{L}^X_{\text{SSL}}, \mathcal{L}^A_{\text{SSL}}, \mathcal{L}^Y_{\text{SSL}}, \mathcal{L}^H_{\text{SSL}}$ for the component representations $\boldsymbol{z}^X_t, \boldsymbol{z}^A_t, \boldsymbol{z}^Y_t$ and the overall representation $\boldsymbol{z}_t$ respectively. The overall self-supervised learning loss is given as:

$$\mathcal{L}_{\text{SSL}} = \mathcal{L}^H_{\text{SSL}} + (\mathcal{L}^X_{\text{SSL}} + \mathcal{L}^A_{\text{SSL}} + \mathcal{L}^Y_{\text{SSL}})/3. \tag{8}$$

**Self-supervised training.** We train our proposed encoder to learn representations of the history with a self-supervised learning framework modified based on MoCo v3 (Chen* et al., 2021) that achieves the state-of-the-art performance in self-supervised vision transformer training. Following MoCo v3, we set up our proposed encoder $f_{\text{enc}}$ as combination of a momentum encoder with the same architecture and initial weights $f^{\text{mo}}_{\text{enc}}$, and a multi-layer perceptron (MLP) as the prediction head $f_{\text{pred\_head}} : \mathbb{R}^{d_{\text{model}}} \to \mathbb{R}^{d_{\text{model}}}$. We first apply the augmentations in (Woo et al., 2022), which includes scaling, shifting and jittering, on each sample in the input batch $\{\bar{\boldsymbol{H}}^{(i)}_t\}^B_{i=1}$ ($B$ is the batch size) and generates the positive sample pair $\{\bar{\boldsymbol{H}}^{'(i)}_t\}^B_{i=1}, \{\bar{\boldsymbol{H}}^{''(i)}_t\}^B_{i=1}$. Their representations are encoded as follows:

$$
\begin{aligned}
\boldsymbol{z}^{'X(i)}_t, \boldsymbol{z}^{'A(i)}_t, \boldsymbol{z}^{'Y(i)}_t, \boldsymbol{z}^{'(i)}_t &= f_{\text{enc}}(\bar{\boldsymbol{H}}^{'(i)}_t), \\
\boldsymbol{z}^{''X(i)}_t, \boldsymbol{z}^{''A(i)}_t, \boldsymbol{z}^{''Y(i)}_t, \boldsymbol{z}^{''(i)}_t &= f_{\text{enc}}(\bar{\boldsymbol{H}}^{''(i)}_t), \\
\boldsymbol{z}^{'X,\text{mo}(i)}_t, \boldsymbol{z}^{'A,\text{mo}(i)}_t, \boldsymbol{z}^{'Y,\text{mo}(i)}_t, \boldsymbol{z}^{'\text{mo}(i)}_t &= f^{\text{mo}}_{\text{enc}}(\bar{\boldsymbol{H}}^{'(i)}_t), \\
\boldsymbol{z}^{''X,\text{mo}(i)}_t, \boldsymbol{z}^{''A,\text{mo}(i)}_t, \boldsymbol{z}^{''Y,\text{mo}(i)}_t, \boldsymbol{z}^{''\text{mo}(i)}_t &= f^{\text{mo}}_{\text{enc}}(\bar{\boldsymbol{H}}^{''(i)}_t).
\end{aligned}
\tag{9}
$$

The vanilla MoCo v3 method adopts the InfoNCE contrastive loss (Oord et al., 2018) as the training objective:

$$
\begin{aligned}
\mathcal{L}^H_{\text{SSL}} =& \mathcal{L}_{\text{InfoNCE}}(\{f_{\text{pred\_head}}(\boldsymbol{z}^{'(i)}_t)\}^B_{i=1}, \{\boldsymbol{z}^{''\text{mo}(i)}_t\}^B_{i=1}) \\
&+ \mathcal{L}_{\text{InfoNCE}}(\{f_{\text{pred\_head}}(\boldsymbol{z}^{''(i)}_t)\}^B_{i=1}, \{\boldsymbol{z}^{'\text{mo}(i)}_t\}^B_{i=1}),
\end{aligned}
\tag{10}
$$

where

$$\mathcal{L}_{\text{InfoNCE}}(\{\boldsymbol{q}^{(i)}\}^B_{i=1}, \{\boldsymbol{k}^{(i)}\}^B_{i=1}) = -\frac{1}{B}\sum^B_{i=1}\log\frac{\exp(\cos\langle\boldsymbol{q}^{(i)}, \boldsymbol{k}^{(i)}\rangle)}{\sum^B_{j=1}\exp(\cos\langle\boldsymbol{q}^{(i)}, \boldsymbol{k}^{(j)}\rangle)}. \tag{11}$$

**Component-wise contrastive loss.** In addition to the contrastive loss of the overall representations, we enhance the training with contrastive losses on each subset of covariates, treatments and outcomes:

$$
\begin{aligned}
\mathcal{L}^{(\cdot)}_{\text{SSL}} =& \mathcal{L}_{\text{InfoNCE}}(\{f_{\text{pred\_head}}(\boldsymbol{z}^{'(\cdot)(i)}_t)\}^B_{i=1}, \{\boldsymbol{z}^{''(\cdot),\text{mo}(i)}_t\}^B_{i=1}) \tag{12} \\
&+ \mathcal{L}_{\text{InfoNCE}}(\{f_{\text{pred\_head}}(\boldsymbol{z}^{''(\cdot)(i)}_t)\}^B_{i=1}, \{\boldsymbol{z}^{'(\cdot),\text{mo}(i)}_t\}^B_{i=1}), \tag{13}
\end{aligned}
$$

where $(\cdot)$ is $X, A, Y$.

## 3.3 NON-AUTOREGRESSIVE OUTCOME PREDICTOR

The architecture of the proposed predictor model is shown in Fig. 2c. At the prediction stage, we first encode the observed history $\bar{\boldsymbol{H}}_t$ with the pretrained encoder, the treatment $\boldsymbol{a}_{t'-1}$ is modeled right before time $t' = t+1, \ldots, t+\tau-1$ with a 1x1 convolution layer, and the remaining treatment sequence $(\boldsymbol{a}_t, \boldsymbol{a}_{t+1}, \ldots, \boldsymbol{a}_{t'-2})$ with a 1D convolution layer. Then, the concatenated encoding is fed into a multi-layer perceptron (MLP) to predict future outcomes $(\hat{\boldsymbol{y}}_{t+1}, \hat{\boldsymbol{y}}_{t+2}, \ldots, \hat{\boldsymbol{y}}_{t+\tau})$. We jointly

train the predictor layers and fine tune the pretrained encoder with the $L_2$ loss of factual outcome estimation weighted for each step:

$$\mathcal{L}_{\text{est}} = \sum_{i=1}^{\tau} w_i \, \|\hat{\boldsymbol{y}}_{t+i} - \boldsymbol{y}_{t+i}\|_2^2 \,, \tag{14}$$

where each $w_i$ is a hyperparameter satisfying $\sum_{i=1}^{\tau} w_i = 1$ and various strategies of setting $w_i$s can be selected via validation errors, which we will discuss in the ablation study (Sec. 4.3).

### 3.4 UNSUPERVISED DOMAIN ADAPTATION VIEW OF COUNTERFACTUAL OUTCOME ESTIMATION

For the given observed history $\bar{\boldsymbol{H}}_t$, treatment sequence $\bar{\boldsymbol{a}}_{t:t+\tau-1}$ to apply, and the outcome $\boldsymbol{y}_{t+\tau}[\boldsymbol{a}_{t:t+\tau-1}](\bar{\boldsymbol{H}}_t)$ to estimate, we notice that learning a counterfactual treatment outcome estimator $f_{\boldsymbol{a}}(\bar{\boldsymbol{H}}_t) = \mathbb{E}(\boldsymbol{y}_{t+\tau}[\boldsymbol{a}]|\bar{\boldsymbol{H}}_t)$ with factual data specifically for a certain treatment sequence $\boldsymbol{a}$ can be viewed as an unsupervised domain adaptation (**UDA**) problem with the treatment value being discrete – any sample in the factual dataset can be categorized into one of the two subsets (1) $\mathcal{S}_{\boldsymbol{a}} = \{(\bar{\boldsymbol{H}}_t^{(i)}, \bar{\boldsymbol{a}}_{t:t+\tau-1}^{(i)}, \boldsymbol{y}_{t+\tau}^{(i)})\}_{\bar{\boldsymbol{a}}_{t:t+\tau-1}^{(i)}=\boldsymbol{a}}$ and (2) $\mathcal{T}_{\bar{\boldsymbol{a}}} = \{(\bar{\boldsymbol{H}}_t^{(i)}, \bar{\boldsymbol{a}}_{t:t+\tau-1}^{(i)}, \boldsymbol{y}_{t+\tau}^{(i)})\}_{\bar{\boldsymbol{a}}_{t:t+\tau-1}^{(i)}\neq\boldsymbol{a}}$. With Assumption B.1, we have $\boldsymbol{y}_{t+\tau}^{(i)} = \boldsymbol{y}_{t+\tau}[\boldsymbol{a}](\bar{\boldsymbol{H}}_t^{(i)})$ in $\mathcal{S}_{\boldsymbol{a}}$ and thus $\boldsymbol{y}_{t+\tau}^{(i)}$ is a label of $f_{\boldsymbol{a}}(\bar{\boldsymbol{H}}_t)$. This does not hold for $\mathcal{T}_{\boldsymbol{a}}$, where $\bar{\boldsymbol{a}}_{t:t+\tau-1}^{(i)} \neq \boldsymbol{a}$. Therefore, $\mathcal{S}_{\boldsymbol{a}}$ and $\mathcal{T}_{\boldsymbol{a}}$ correspond to the **labeled** and the **unlabeled** dataset in UDA. Considering the existence of treatment bias, $P_{\mathcal{T}_{\boldsymbol{a}}}(\bar{\boldsymbol{H}}_t) = P(\bar{\boldsymbol{H}}_t|\bar{\boldsymbol{a}}_{t:t+\tau-1} \neq \boldsymbol{a}) \neq P(\bar{\boldsymbol{H}}_t|\bar{\boldsymbol{a}}_{t:t+\tau-1} = \boldsymbol{a}) = P_{\mathcal{S}_{\boldsymbol{a}}}(\bar{\boldsymbol{H}}_t)$, which corresponds to the distribution shift between the labeled source domain and the unlabeled target domain in UDA. Notice that the source/target domains here are used for describing the labeled and unlabeled subsets regarding a treatment value to help us analyze the error of counterfactual outcome estimation in UDA framework, and are different from the definitions we use in Section 2. In the latter case, source/target domains describe the different distributions of $\bar{\boldsymbol{H}}_t$ between train/test data. As a natural generalization, we analyze the upper bound of contrastive learning for counterfactual outcome estimation based on the transferability analysis of contrastive learning in UDA (HaoChen et al., 2022):

**Theorem 3.1** (Upper bound of counterfactual outcome estimator). *Suppose that Assumptions D.2, D.3, and D.4 hold for the set of observed history $\mathcal{H}$ and its positive-pair graph $G(\mathcal{H}, w)$, and the representation dimension $k \geq 2m$. Let $r$ be a minimizer of the generalized spectral contrastive loss on factual data and the regression head $f_{\boldsymbol{a}}$ be constructed in Alg. 1 with labeled data. We have*

$$\mathcal{E}_{\mathcal{D}}(f_{\boldsymbol{a}}) \lesssim P(\boldsymbol{a})\mathcal{E}_{\mathcal{S}_{\boldsymbol{a}}}(f_{\boldsymbol{a}}) + (1 - P(\boldsymbol{a})) \left[ \epsilon^2 + (4B^2 - \epsilon^2)\frac{r}{\alpha^2\gamma^4} \cdot \exp(-\Omega(\frac{\rho\gamma^2}{\alpha^2})) \right], \tag{15}$$

where $P(\boldsymbol{a})$ is the prior probability of the treatment $\boldsymbol{a}$ to apply. $\mathcal{E}_{\mathcal{S}_{\boldsymbol{a}}}(f_{\boldsymbol{a}})$ is the outcome estimation error of $f_{\boldsymbol{a}}$ in $\mathcal{S}_{\boldsymbol{a}}$, which can be optimized with supervised learning. $\alpha, r, \gamma, \rho$ are parameters in Assumptions D.2, D.3, D.4. $\epsilon$ is the hyperparameter in Alg. 1. $B$ is the upper bound of outcome and predicted outcome values[2]. When $\gamma \geq \alpha^{1/2}$ and $\rho$ is comparable to $\alpha$, $\rho\gamma^2 \gg \alpha^2$ and lead to a small upper bound. We provide the proof in Appendix D.

## 4 EXPERIMENTS

**Datasets.** Following the common evaluation setup for counterfactual treatment outcome estimation overtime in (Lim, 2018; Bica et al., 2020; Melnychuk et al., 2022), we use datasets from both synthetic simulation and real-world observation in our experiments. We provide more detailed dataset description in Appendix E. **(1) Tumor growth.** Following previous work (Lim, 2018; Bica et al., 2020; Melnychuk et al., 2022), we run the pharmacokinetic-pharmacodynamic(PK-PD) tumor growth simulation and generates a fully-synthetic dataset. The PK-PD simulation (Geng et al., 2017) is a state-of-the-art bio-mathematical model simulating the combined effect of chemotherapy and radiotherapy on tumor volumes. **(2) Semi-synthetic MIMIC-III.** (Melnychuk et al., 2022) constructs

---

[2]Bounded outcome values can be achieved through normalization.

a semi-synthetic dataset by simulating outcomes under endogenous dependencies on time and exogenous dependencies on observational patient trajectories that are high dimensional and contain long-range dependencies. We include it in our evaluation as a more challenging synthetic dataset. **(3) M5.** The M5 Forecasting dataset (Makridakis et al., 2022) contains daily sales of Walmart stores across three US states, along with the metadata of items and stores, as well as explanatory variables such as price and special events. We transform it to treatment outcome estimation task with the treatment variable of item price, and the outcome variable of the sales of items. Covariate variables include all remaining features. With synthetic data, we report the counterfactual outcome estimation errors and compare the performance of SCOT with baselines. However, with real-world data, the counterfactual outcome cannot be observed or simulated, thus, we only report the prediction errors of factual outcome.

**Feature distribution shifts.** To achieve a comprehensive evaluation of counterfactual outcome estimation performance, we introduce feature distribution shifts into the datasets. For each dataset, we split based on static characteristics of subjects into a subset in the source domain and a subset in the target domain. Each subset is further divided into train/validation/test sets. We summarize the main statistics of datasets in Table 4 in the appendix.

**Baselines.** We select comprehensive methods for estimating counterfactual outcomes over time as baselines, including MSM (Robins et al., 2000), RMSN (Lim, 2018), CRN (Bica et al., 2020), G-Net (Li et al., 2021), and Causal Transformer (CT) (Melnychuk et al., 2022). MSM has difficulty converging when trained with high-dimensional input in semi-synthetic MIMIC-III and M5 datasets and we thus only evaluate it for tumor growth. We empirically find that the balanced representation training losses proposed in CRN and CT do not bring a robust improvement over their variants trained only with the empirical risk minimization (ERM) on factual outcomes. Therefore, we also include these variants, CRN(ERM) and CT(ERM), as baselines.

## 4.1 ZERO-SHOT TRANSFER SETUP

To showcase cold-start prediction capabilities, in this setup, we focus on the performance on the target domain after training the model in the source domain, with distributional difference in features. Results are shown in Table 1. SCOT demonstrates the state-of-the-art performance in a majority of horizons across datasets (4/6 in Tumor growth, 6/6 in Semi-synthetic MIMIC-III and M5). On average, SCOT decreases the outcome estimation errors by over 6.2%, 22.5% and 26.3% compared to baselines. Results demonstrate the strong transferability of SCOT in the zero-shot transfer setting.

Table 1: Results of zero-shot transfer setup for multi-step outcome estimation. We report the mean $\pm$ standard deviation of Rooted Mean Squared Errors (RMSEs $\downarrow$) over 5 runs. **Bold**: the best results. Underline: the 2nd best results.

| Dataset | Method | $\tau = 1$ | $\tau = 2$ | $\tau = 3$ | $\tau = 4$ | $\tau = 5$ | $\tau = 6$ | Avg | Gain(%) |
|---|---|---|---|---|---|---|---|---|---|
| Tumor growth | MSM | $1.0515_{\pm 0.0674}$ | $\mathbf{0.5048_{\pm 0.0591}}$ | $\mathbf{0.7583_{\pm 0.0831}}$ | $0.9685_{\pm 0.1066}$ | $1.1561_{\pm 0.1243}$ | $1.3372_{\pm 0.1356}$ | $0.9627_{\pm 0.0923}$ | 6.2% |
| | RMSN | $1.2406_{\pm 0.1301}$ | $1.0914_{\pm 0.0346}$ | $1.1315_{\pm 0.0634}$ | $1.1583_{\pm 0.0810}$ | $1.1674_{\pm 0.0913}$ | $\underline{1.1531_{\pm 0.0919}}$ | $1.1571_{\pm 0.0660}$ | 22.0% |
| | CRN(ERM) | $1.2924_{\pm 0.0772}$ | $1.1769_{\pm 0.1058}$ | $1.1728_{\pm 0.1136}$ | $1.1906_{\pm 0.1106}$ | $1.1997_{\pm 0.1061}$ | $\underline{1.1883_{\pm 0.0985}}$ | $1.2035_{\pm 0.0901}$ | 25.0% |
| | CRN | $1.5643_{\pm 0.0667}$ | $10.1978_{\pm 14.9899}$ | $14.5460_{\pm 23.3314}$ | $17.2080_{\pm 28.4348}$ | $18.8712_{\pm 31.5756}$ | $19.9238_{\pm 33.5369}$ | $13.7185_{\pm 21.9748}$ | 93.4% |
| | CT(ERM) | $0.9729_{\pm 0.0718}$ | $1.0217_{\pm 0.0292}$ | $1.1173_{\pm 0.0457}$ | $1.1904_{\pm 0.0395}$ | $1.2359_{\pm 0.0618}$ | $1.2913_{\pm 0.0939}$ | $1.1383_{\pm 0.0251}$ | 20.7% |
| | CT | $1.0272_{\pm 0.1077}$ | $1.1428_{\pm 0.2182}$ | $1.2708_{\pm 0.2471}$ | $1.3608_{\pm 0.2681}$ | $1.4166_{\pm 0.2935}$ | $1.4322_{\pm 0.3138}$ | $1.2751_{\pm 0.2326}$ | 29.2% |
| | G-Net | $1.0492_{\pm 0.0529}$ | $1.0125_{\pm 0.0767}$ | $1.1271_{\pm 0.0876}$ | $1.2153_{\pm 0.0777}$ | $1.2549_{\pm 0.0727}$ | $1.2543_{\pm 0.0678}$ | $1.1522_{\pm 0.0537}$ | 21.7% |
| | SCOT | $\mathbf{0.8767_{\pm 0.0492}}$ | $\underline{0.7995_{\pm 0.0853}}$ | $\underline{0.8282_{\pm 0.0801}}$ | $\mathbf{0.9021_{\pm 0.1062}}$ | $\mathbf{0.9888_{\pm 0.1280}}$ | $\mathbf{1.0210_{\pm 0.1168}}$ | $\mathbf{0.9027_{\pm 0.0814}}$ | (-) |
| Semi-synthetic MIMIC-III | RMSN | $0.2551_{\pm 0.0303}$ | $0.6641_{\pm 0.1092}$ | $0.9107_{\pm 0.1915}$ | $1.1217_{\pm 0.2916}$ | $1.2821_{\pm 0.3603}$ | $1.3950_{\pm 0.4038}$ | $0.9381_{\pm 0.2210}$ | 44.7% |
| | CRN(ERM) | $\underline{0.2506_{\pm 0.0303}}$ | $0.5545_{\pm 0.0917}$ | $0.7581_{\pm 0.1112}$ | $0.9018_{\pm 0.1547}$ | $1.0113_{\pm 0.1941}$ | $1.1068_{\pm 0.2324}$ | $0.7639_{\pm 0.1238}$ | 32.1% |
| | CRN | $0.4041_{\pm 0.0537}$ | $0.8256_{\pm 0.1767}$ | $1.0439_{\pm 0.1958}$ | $1.1807_{\pm 0.1725}$ | $1.3121_{\pm 0.2229}$ | $1.4374_{\pm 0.3089}$ | $1.0340_{\pm 0.1606}$ | 49.8% |
| | CT(ERM) | $0.2762_{\pm 0.0804}$ | $\underline{0.5397_{\pm 0.1181}}$ | $0.6765_{\pm 0.1417}$ | $0.7728_{\pm 0.1636}$ | $0.8451_{\pm 0.1850}$ | $0.9028_{\pm 0.2070}$ | $0.6688_{\pm 0.1472}$ | 22.5% |
| | CT | $0.3138_{\pm 0.0458}$ | $0.5992_{\pm 0.0492}$ | $0.7576_{\pm 0.0694}$ | $0.8695_{\pm 0.0921}$ | $0.9510_{\pm 0.1118}$ | $1.0128_{\pm 0.1274}$ | $0.7506_{\pm 0.0797}$ | 30.9% |
| | G-Net | $0.5514_{\pm 0.1502}$ | $0.9398_{\pm 0.2384}$ | $1.2461_{\pm 0.3321}$ | $1.4985_{\pm 0.4024}$ | $1.7045_{\pm 0.4463}$ | $1.8731_{\pm 0.4660}$ | $1.3022_{\pm 0.3367}$ | 60.2% |
| | SCOT | $\mathbf{0.2266_{\pm 0.0249}}$ | $\mathbf{0.4501_{\pm 0.0893}}$ | $\mathbf{0.5406_{\pm 0.0987}}$ | $\mathbf{0.5964_{\pm 0.1020}}$ | $\mathbf{0.6344_{\pm 0.1040}}$ | $\mathbf{0.6637_{\pm 0.1052}}$ | $\mathbf{0.5186_{\pm 0.0869}}$ | (-) |
| M5 | RMSN | $15.1616_{\pm 2.0027}$ | $13.9966_{\pm 0.5316}$ | $13.4899_{\pm 1.2632}$ | $13.5162_{\pm 1.7437}$ | $13.8004_{\pm 2.0637}$ | $14.3366_{\pm 2.3891}$ | $13.8280_{\pm 1.5526}$ | 47.5% |
| | CRN(ERM) | $9.8859_{\pm 1.2980}$ | $20.8199_{\pm 3.9049}$ | $38.2653_{\pm 8.9897}$ | $59.4192_{\pm 16.2788}$ | $82.9515_{\pm 26.1928}$ | $105.8120_{\pm 35.5325}$ | $61.4536_{\pm 17.5760}$ | 88.2% |
| | CRN | $8.1119_{\pm 0.3183}$ | $10.3741_{\pm 2.2616}$ | $12.9356_{\pm 3.1588}$ | $15.4168_{\pm 3.8002}$ | $18.1382_{\pm 4.6750}$ | $21.1337_{\pm 5.4694}$ | $15.5997_{\pm 3.7687}$ | 53.5% |
| | CT(ERM) | $7.1253_{\pm 0.5777}$ | $8.3438_{\pm 1.0313}$ | $9.2014_{\pm 1.4146}$ | $9.9409_{\pm 1.7572}$ | $10.6726_{\pm 2.1718}$ | $11.3597_{\pm 2.5966}$ | $9.9037_{\pm 1.7852}$ | 26.7% |
| | CT | $\underline{7.1239_{\pm 0.5770}}$ | $8.2939_{\pm 0.9702}$ | $9.1465_{\pm 1.3397}$ | $9.9091_{\pm 1.7198}$ | $\underline{10.6311_{\pm 2.0328}}$ | $\underline{11.3032_{\pm 2.4185}}$ | $9.8568_{\pm 1.6959}$ | 26.3% |
| | G-Net | $7.5358_{\pm 0.1605}$ | $8.6077_{\pm 0.3166}$ | $9.7167_{\pm 0.4861}$ | $10.8993_{\pm 0.6902}$ | $12.3477_{\pm 0.8940}$ | $13.8200_{\pm 1.1193}$ | $10.4879_{\pm 0.6078}$ | 30.8% |
| | SCOT | $\mathbf{6.4054_{\pm 0.0547}}$ | $\mathbf{6.9328_{\pm 0.0634}}$ | $\mathbf{7.2428_{\pm 0.0700}}$ | $\mathbf{7.4585_{\pm 0.0580}}$ | $\mathbf{7.7012_{\pm 0.0627}}$ | $\mathbf{7.8278_{\pm 0.0651}}$ | $\mathbf{7.2614_{\pm 0.0609}}$ | (-) |

## 4.2 DATA-EFFICIENT TRANSFER LEARNING SETUP

Effectively utilizing small amount of target domain data can be important, and we showcase that it is indeed one of the key strengths of the proposed approach.

To demonstrate this, for the Tumor Growth Dataset, we fine-tune each method trained on the source domain with 100 sequences from the target domain. For the semi-synthetic MIMIC-III and M5 datasets, we set the number of target domain samples for fine-tuning to be 10% of the number of samples of the target domain in the original dataset. To achieve a fair comparison, we fine-tune each method until it reaches the lowest factual outcome estimation error on a separate validation set in the target domain.

Table 2 compares the performance of all methods in data-efficient transfer learning setup. For the majority of horizons (4/6 in Tumor growth, 5/6 in Semi-synthetic MIMIC-III and M5), we observe that SCOT achieves the state-of-the-art performance after fine-tuning. Again, SCOT reduces the outcome estimation errors by at least 7.8%, 9.9% and 4.11% in the three datasets respectively.

Table 2: Results of the data-efficient transfer learning setup for multi-step outcome estimation. We report the mean $\pm$ standard deviation of Rooted Mean Squared Errors (RMSEs $\downarrow$) over 5 runs. **Bold**: the best results. Underline: the 2nd best results.

| Dataset | Method | $\tau=1$ | $\tau=2$ | $\tau=3$ | $\tau=4$ | $\tau=5$ | $\tau=6$ | Avg | Gain(%) |
|---|---|---|---|---|---|---|---|---|---|
| Tumor growth | MSM | $1.0436_{\pm 0.0671}$ | $\mathbf{0.5023_{\pm 0.0588}}$ | $\mathbf{0.7475_{\pm 0.0829}}$ | $0.9537_{\pm 0.1060}$ | $1.1376_{\pm 0.1233}$ | $1.3146_{\pm 0.1338}$ | $0.9499_{\pm 0.0915}$ | 7.8% |
| | RMSN | $1.1839_{\pm 0.0842}$ | $1.0912_{\pm 0.0405}$ | $1.1215_{\pm 0.0593}$ | $1.1538_{\pm 0.0688}$ | $1.1728_{\pm 0.0773}$ | $1.1740_{\pm 0.0830}$ | $1.1495_{\pm 0.0529}$ | 23.8% |
| | CRN(ERM) | $1.2648_{\pm 0.0689}$ | $1.1740_{\pm 0.1015}$ | $1.1507_{\pm 0.1016}$ | $1.1474_{\pm 0.1070}$ | $1.1414_{\pm 0.1041}$ | $1.1203_{\pm 0.0906}$ | $1.1664_{\pm 0.0786}$ | 24.9% |
| | CRN | $1.4721_{\pm 0.0620}$ | $8.4730_{\pm 13.1786}$ | $13.1474_{\pm 21.7125}$ | $17.2385_{\pm 29.3990}$ | $21.5399_{\pm 37.7228}$ | $25.3391_{\pm 45.1447}$ | $14.5350_{\pm 24.5243}$ | 94.0% |
| | CT(ERM) | $0.8947_{\pm 0.0668}$ | $0.8700_{\pm 0.0857}$ | $0.9507_{\pm 0.1309}$ | $1.0031_{\pm 0.1502}$ | $\underline{1.0371_{\pm 0.1545}}$ | $\underline{1.0668_{\pm 0.1565}}$ | $0.9704_{\pm 0.1098}$ | 9.7% |
| | CT | $0.9545_{\pm 0.0782}$ | $0.9494_{\pm 0.1597}$ | $1.0225_{\pm 0.1562}$ | $1.1062_{\pm 0.1377}$ | $1.1455_{\pm 0.1192}$ | $1.1562_{\pm 0.0953}$ | $1.0557_{\pm 0.1136}$ | 17.0% |
| | G-Net | $1.0335_{\pm 0.0622}$ | $1.0154_{\pm 0.1100}$ | $1.1105_{\pm 0.1476}$ | $1.1859_{\pm 0.1620}$ | $1.2257_{\pm 0.1693}$ | $1.2198_{\pm 0.1508}$ | $1.1318_{\pm 0.1118}$ | 22.6% |
| | SCOT | $\mathbf{0.8654_{\pm 0.0328}}$ | $\underline{0.7945_{\pm 0.0532}}$ | $\underline{0.8248_{\pm 0.0751}}$ | $\mathbf{0.8754_{\pm 0.0987}}$ | $\mathbf{0.9378_{\pm 0.1176}}$ | $\mathbf{0.9594_{\pm 0.1062}}$ | $\mathbf{0.8762_{\pm 0.0720}}$ | (-) |
| Semi-synthetic MIMIC-III | RMSN | $0.2100_{\pm 0.0192}$ | $0.6084_{\pm 0.1114}$ | $0.7745_{\pm 0.1180}$ | $0.8908_{\pm 0.1402}$ | $0.9776_{\pm 0.1505}$ | $1.0440_{\pm 0.1529}$ | $0.7509_{\pm 0.1123}$ | 31.0% |
| | CRN(ERM) | $\mathbf{0.1946_{\pm 0.0158}}$ | $0.4770_{\pm 0.0808}$ | $0.5983_{\pm 0.0923}$ | $0.6786_{\pm 0.1004}$ | $0.7315_{\pm 0.1047}$ | $0.7690_{\pm 0.1070}$ | $0.5748_{\pm 0.0823}$ | 9.9% |
| | CRN | $0.2955_{\pm 0.0256}$ | $0.5051_{\pm 0.0748}$ | $0.6361_{\pm 0.0786}$ | $0.7277_{\pm 0.0783}$ | $0.7919_{\pm 0.0764}$ | $0.8379_{\pm 0.0759}$ | $0.6324_{\pm 0.0656}$ | 18.1% |
| | CT(ERM) | $0.2704_{\pm 0.0631}$ | $0.5347_{\pm 0.1061}$ | $0.6712_{\pm 0.1252}$ | $0.7679_{\pm 0.1433}$ | $0.8402_{\pm 0.1607}$ | $0.8968_{\pm 0.1784}$ | $0.6635_{\pm 0.1279}$ | 21.9% |
| | CT | $0.3105_{\pm 0.0459}$ | $0.5840_{\pm 0.0633}$ | $0.7414_{\pm 0.0887}$ | $0.8530_{\pm 0.1157}$ | $0.9348_{\pm 0.1392}$ | $0.9974_{\pm 0.1608}$ | $0.7368_{\pm 0.0971}$ | 29.7% |
| | G-Net | $0.3814_{\pm 0.0556}$ | $0.6519_{\pm 0.0856}$ | $0.8183_{\pm 0.1122}$ | $0.9413_{\pm 0.1365}$ | $1.0359_{\pm 0.1592}$ | $1.1117_{\pm 0.1795}$ | $0.8234_{\pm 0.1191}$ | 37.1% |
| | SCOT | $0.2288_{\pm 0.0229}$ | $\mathbf{0.4496_{\pm 0.0877}}$ | $\mathbf{0.5393_{\pm 0.0962}}$ | $\mathbf{0.5946_{\pm 0.0990}}$ | $\mathbf{0.6326_{\pm 0.1013}}$ | $\mathbf{0.6626_{\pm 0.1026}}$ | $\mathbf{0.5179_{\pm 0.0844}}$ | (-) |
| M5 | RMSN | $13.9705_{\pm 0.3867}$ | $13.6233_{\pm 0.8150}$ | $13.3291_{\pm 1.2900}$ | $13.1984_{\pm 1.3892}$ | $13.0889_{\pm 1.2605}$ | $13.0108_{\pm 1.1173}$ | $13.2501_{\pm 1.1696}$ | 45.92% |
| | CRN(ERM) | $6.3558_{\pm 0.0594}$ | $7.0530_{\pm 0.0433}$ | $7.3452_{\pm 0.0447}$ | $7.5541_{\pm 0.0392}$ | $7.7636_{\pm 0.0450}$ | $7.9247_{\pm 0.0561}$ | $7.5281_{\pm 0.0447}$ | 4.82% |
| | CRN | $6.2868_{\pm 0.0471}$ | $7.0282_{\pm 0.0482}$ | $7.3327_{\pm 0.0610}$ | $\underline{7.5378_{\pm 0.0521}}$ | $\underline{7.7492_{\pm 0.0586}}$ | $\underline{7.9094_{\pm 0.0676}}$ | $7.5115_{\pm 0.0572}$ | 4.60% |
| | CT(ERM) | $\mathbf{6.1720_{\pm 0.0354}}$ | $6.9309_{\pm 0.0571}$ | $7.2855_{\pm 0.0889}$ | $7.5418_{\pm 0.1191}$ | $7.7839_{\pm 0.1283}$ | $7.9425_{\pm 0.1430}$ | $7.4969_{\pm 0.1058}$ | 4.42% |
| | CT | $6.2041_{\pm 0.0252}$ | $7.0022_{\pm 0.0372}$ | $7.3675_{\pm 0.0513}$ | $7.6394_{\pm 0.0894}$ | $7.8932_{\pm 0.1153}$ | $8.0701_{\pm 0.1456}$ | $7.5945_{\pm 0.0845}$ | 5.65% |
| | G-Net | $6.7077_{\pm 0.1006}$ | $7.0479_{\pm 0.1069}$ | $7.3872_{\pm 0.1349}$ | $7.6545_{\pm 0.1596}$ | $7.9188_{\pm 0.1800}$ | $8.1186_{\pm 0.2058}$ | $\underline{7.4725_{\pm 0.1461}}$ | 4.11% |
| | SCOT | $6.3026_{\pm 0.0519}$ | $\mathbf{6.8364_{\pm 0.0560}}$ | $\mathbf{7.1464_{\pm 0.0674}}$ | $\mathbf{7.3634_{\pm 0.0619}}$ | $\mathbf{7.6058_{\pm 0.0640}}$ | $\mathbf{7.7393_{\pm 0.0637}}$ | $\mathbf{7.1656_{\pm 0.0592}}$ | (-) |

## 4.3 ABLATION STUDIES

Table 3: Ablation studies for multi-step outcome estimation. We report the mean $\pm$ standard deviation of Rooted Mean Squared Errors (RMSEs $\downarrow$) over 5 runs. **Bold**: the best results.

| Dataset | Component | Choice | $\tau=1$ | $\tau=2$ | $\tau=3$ | $\tau=4$ | $\tau=5$ | $\tau=6$ | Avg | Gain(%) |
|---|---|---|---|---|---|---|---|---|---|---|
| Semi-synthetic MIMIC-III | SCOT | | $0.2266_{\pm 0.0249}$ | $0.4501_{\pm 0.0893}$ | $\mathbf{0.5406_{\pm 0.0987}}$ | $\mathbf{0.5964_{\pm 0.1020}}$ | $\mathbf{0.6344_{\pm 0.1040}}$ | $0.6637_{\pm 0.1052}$ | $0.5186_{\pm 0.0869}$ | (-) |
| | Encoder | w/ VT | $0.4897_{\pm 0.0888}$ | $0.6161_{\pm 0.1139}$ | $0.6978_{\pm 0.1200}$ | $0.7428_{\pm 0.1217}$ | $0.7705_{\pm 0.1196}$ | $0.7910_{\pm 0.1182}$ | $0.6846_{\pm 0.1130}$ | 24.2% |
| | | w/ CT | $0.3519_{\pm 0.0584}$ | $0.4936_{\pm 0.0897}$ | $0.5762_{\pm 0.0967}$ | $0.6279_{\pm 0.1017}$ | $0.6634_{\pm 0.1045}$ | $0.6874_{\pm 0.1029}$ | $0.5667_{\pm 0.0919}$ | 8.5% |
| | FPE | w/ abs | $0.2981_{\pm 0.0444}$ | $0.4679_{\pm 0.0940}$ | $0.5561_{\pm 0.1050}$ | $0.6091_{\pm 0.1079}$ | $0.6446_{\pm 0.1115}$ | $0.6694_{\pm 0.1128}$ | $0.5409_{\pm 0.0951}$ | 4.1% |
| | SSL Loss | none | $0.2998_{\pm 0.0466}$ | $0.4718_{\pm 0.0905}$ | $0.5579_{\pm 0.1007}$ | $0.6117_{\pm 0.1035}$ | $0.6460_{\pm 0.1060}$ | $0.6680_{\pm 0.1060}$ | $0.5425_{\pm 0.0914}$ | 4.4% |
| | | w/o comp | $0.2884_{\pm 0.0421}$ | $0.4603_{\pm 0.0898}$ | $0.5475_{\pm 0.1032}$ | $0.6013_{\pm 0.1077}$ | $0.6353_{\pm 0.1079}$ | $\mathbf{0.6610_{\pm 0.1075}}$ | $0.5323_{\pm 0.0926}$ | 2.6% |
| | SupL Loss | w/ uni | $0.2910_{\pm 0.0355}$ | $0.4656_{\pm 0.0873}$ | $0.5547_{\pm 0.0998}$ | $0.6094_{\pm 0.1039}$ | $0.6440_{\pm 0.1059}$ | $0.6681_{\pm 0.1070}$ | $0.5884_{\pm 0.1006}$ | 11.9% |
| | | w/ sq.inv. | $\mathbf{0.1968_{\pm 0.0148}}$ | $\mathbf{0.4456_{\pm 0.0815}}$ | $0.5415_{\pm 0.0906}$ | $0.6021_{\pm 0.0954}$ | $0.6431_{\pm 0.0956}$ | $0.6761_{\pm 0.0948}$ | $\mathbf{0.5175_{\pm 0.0780}}$ | -0.2% |
| | Decoder | w/ autoreg | $0.2049_{\pm 0.0118}$ | $0.7036_{\pm 0.1422}$ | $1.0234_{\pm 0.2214}$ | $1.2023_{\pm 0.2745}$ | $1.4692_{\pm 0.3711}$ | $1.6577_{\pm 0.4959}$ | $1.0435_{\pm 0.2437}$ | 50.3% |
| M5 | SCOT | | $6.4054_{\pm 0.0547}$ | $\mathbf{6.9328_{\pm 0.0634}}$ | $\mathbf{7.2428_{\pm 0.0700}}$ | $\mathbf{7.4585_{\pm 0.0580}}$ | $\mathbf{7.7012_{\pm 0.0627}}$ | $\mathbf{7.8278_{\pm 0.0651}}$ | $\mathbf{7.2614_{\pm 0.0609}}$ | (-) |
| | Encoder | w/ VT | $17.8226_{\pm 4.7807}$ | $17.6769_{\pm 4.5561}$ | $17.5937_{\pm 4.4219}$ | $17.5279_{\pm 4.2936}$ | $17.4113_{\pm 4.1662}$ | $17.2706_{\pm 4.0453}$ | $17.5505_{\pm 4.3765}$ | 58.6% |
| | | w/ CT | $6.7386_{\pm 0.2326}$ | $7.1911_{\pm 0.2474}$ | $7.4549_{\pm 0.2322}$ | $7.6524_{\pm 0.2061}$ | $7.8488_{\pm 0.2021}$ | $7.9589_{\pm 0.2006}$ | $7.4741_{\pm 0.2176}$ | 2.8% |
| | FPE | w/ abs | $6.4089_{\pm 0.0693}$ | $6.9648_{\pm 0.0617}$ | $7.2776_{\pm 0.0528}$ | $7.4834_{\pm 0.0430}$ | $7.7214_{\pm 0.0421}$ | $7.8479_{\pm 0.0370}$ | $7.2840_{\pm 0.0486}$ | 0.3% |
| | SSL Loss | none | $6.4296_{\pm 0.1193}$ | $6.9434_{\pm 0.0796}$ | $7.2548_{\pm 0.0699}$ | $7.4744_{\pm 0.0753}$ | $7.7117_{\pm 0.0728}$ | $7.8429_{\pm 0.0817}$ | $7.2761_{\pm 0.0827}$ | 0.2% |
| | | w/o comp | $6.4637_{\pm 0.0926}$ | $6.9847_{\pm 0.0764}$ | $7.2934_{\pm 0.0705}$ | $7.5093_{\pm 0.0661}$ | $7.7497_{\pm 0.0669}$ | $7.8748_{\pm 0.0608}$ | $7.3126_{\pm 0.0715}$ | 0.7% |
| | SupL Loss | w/ sq.inv. | $\mathbf{6.3425_{\pm 0.0461}}$ | $6.9760_{\pm 0.0523}$ | $7.3170_{\pm 0.0538}$ | $7.5366_{\pm 0.0614}$ | $7.8015_{\pm 0.0658}$ | $7.9439_{\pm 0.0736}$ | $7.3196_{\pm 0.0583}$ | 0.8% |
| | | w/ inv | $6.3575_{\pm 0.0473}$ | $6.9427_{\pm 0.0381}$ | $7.2766_{\pm 0.0422}$ | $7.4932_{\pm 0.0447}$ | $7.7431_{\pm 0.0531}$ | $7.8785_{\pm 0.0628}$ | $7.2819_{\pm 0.0464}$ | 0.3% |
| | Decoder | w/ autoreg | $6.3572_{\pm 0.0621}$ | $>20$ | $>20$ | $>20$ | $>20$ | $>20$ | $>20$ | $>60\%$ |

We conduct ablation studies in the zero-shot transfer setup to validate the design of SCOT. We choose the feature-rich datasets: semi-synthetic MIMIC-III and M5 since they contain complex dynamics and thus are more viable for evaluating components capturing temporal and feature-wise interactions.

**Encoder.** To validate the impact of our proposed encoder architecture, we replace the it with two variants: (1) Vanilla Transformer (**w/ VT**). A vanilla transformer with temporal causal attention, which takes the history with all features concatenated as multivariate time series input. (2) CT (**w/ CT**). The encoder architecture proposed by Causal Transformer (Melnychuk et al., 2022) that concatenates features grouped by covariates/treatments/outcomes into 3 subsets first, then applies self-attention/cross-attention among sequences with each group of features/each pair of feature groups in an alternating way. Rows "Backbone | w/VT(w/CT)" in Table 3 demonstrate the superior performance of our proposed encoder architecture. We observe that both methods (CT, SCOT) processing features respectively outperform VT that simply concatenates all features, marking the importance of explicitly modeling feature interactions. Moreover, the finer-grained modeling of feature interactions between each pair of features in SCOT further improves the estimation performance compared to the coarser modeling of interactions between feature subsets in CT.

**Feature positional encoding (FPE).** We replace the tree-based feature positional encoding with its absolute variant (**w/abs**): each feature maps to a separate learnable encoding vector. We observe that the tree-based positional encoding has gains of 4.1% and 0.3% over the absolute variant in the two datasets respectively.

**Self-supervised loss (SSL).** To validate the improvement brought by introducing self-supervised learning as well as the choice of its training loss, we compare SCOT with two variants: (i) **none.** A model with the same architecture as SCOT but trained with factual estimation losses only; and (ii) **w/o comp.** with vanilla MoCo v3 training loss in Eq. 10 for self-supervised learning. Rows "SSL Loss | none(w/o comp)" in Table 3 compare the estimation performance of the aforementioned choices of self-supervised learning losses and validate the effectiveness of our component-wise contrastive loss in self-supervised learning.

**Supervised loss (SupL).** We consider different choices of the hyperparameter in the supervised training loss of Eq. 14: (1) **w/ uni**: a uniform weight with each $w_i = 1/\tau$; (2) **w/ inv**: weights in proportion to the inverse of horizon $w_i = \frac{1/i}{\sum_{j=1}^{\tau} 1/j}$; (3) **w/ sq.inv.**: weights in proportion to the inverse of the squared horizon $w_i = \frac{1/i^2}{\sum_{j=1}^{\tau} 1/j^2}$. Both (2) and (3) are designed to enhance the short-term outcome estimation performance. We select the weights by validation error for each dataset (w/inv for Semi-synthetic MIMIC-III and w/uni for M5), and compare it to the other two variants. While the relative performance order varies across datasets, all variants can outperform the best baseline results in Table 1.

**Decoder.** We validate the effectiveness of our non-autoregressive design of the decoder and compare it with an autoregressive alternative (**w/ autoreg**) by including the previous outcome in input features. While results in rows "Decoder | w/ autoreg" show good performance in very short horizons ($\tau = 1$), multistep outcome estimation errors quickly diverges with the horizon increasing.

## 5 CONCLUSION

In this work, we propose a self-supervised learning framework - Self-supervised Counterfactual Transformer - to tackle the challenges associated with accurately estimating treatment outcomes over time using observed history, which is a crucial component in areas where randomized controlled trials (RCTs) are not feasible. By integrating self-supervised learning and the Transformer-based encoder combining temporal with feature-wise attention, we've achieved notable advances in estimation accuracy and cross-domain generalization performance.

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

## A  RELATED WORK

**Counterfactual treatment outcome estimation over time.**  Early works in counterfactual treatment outcome estimation were first developed for epidemiology and can be considered under 3 major groups: G-computation, marginal structural models (MSMs), and structural nested models (Robins, 1986; 1994; Robins et al., 2000; Robins & Hernan, 2008). One major shortcoming of these is that they are built on linear models and suffer from the limited model capacity when facing time series data with complex temporal dependencies. Follow-up works address the limitation in expressiveness with Bayesian non-parametric methods (Xu et al., 2016; Soleimani et al., 2017; Schulam & Saria, 2017) or more expressive deep neural networks (DNNs) such as recurrent neural networks (RNNs). For example, recurrent marginal structural networks (RMSNs) (Lim, 2018) replace the linear model in MSM with an RNN-based architecture for forecasting treatment outcomes. G-Net (Li et al., 2021) also adopts RNN instead of classical regression models in the g-computation framework. Inspired by the success of representation learning for domain adaptation and generalization (Ganin et al., 2016; Tzeng et al., 2015), more recent works explore learning representations that are both predictive for outcome estimation and balanced regardless of the treatment bias in training data. Counterfactual recurrent network (CRN) (Bica et al., 2020) trains an RNN-based model with both the factual outcome regression loss and the gradient reversal (Ganin et al., 2016) w.r.t. the treatment prediction loss. The former loss encourages the learned representations to be predictive of outcomes while the latter encourages the representations to be homogeneous given different treatments. The joint training target leads to informative and balanced representations. With similar motivations, (Melnychuk et al., 2022) replaces the RNN-based architecture with a Transformer-based (Vaswani et al., 2017) one along with the domain confusion loss (Tzeng et al., 2015) to learn treatment-agnostic representations. Given the flexibility of the choice of model architectures, recent works extend temporal counterfactual outcome estimation to irregular time series (Seedat et al., 2022; Cao et al., 2023), temporal point process (Zhang et al., 2022b), and graph-structured spatiotemporal data (Jiang et al., 2023) with the help of (Kidger et al., 2020) and (Huang et al., 2020). While existing works claim that both predictive and balanced representations are critical in accurate counterfactual outcome estimation, we empirically find that the impact of representation balancing is inconsistent and marginal. In contrast, improving the expressiveness of representations brings more robust improvements. Our main contributions over existing works are as follows: (1) As far as we know, our work is the first to adapt self-supervised learning (SSL) in temporal counterfactual outcome estimation. Instead of simply putting SSL in our task, we also provide theoretical insights for the contribution of SSL. (2) In addition to SSL, we also propose a new encoder architecture combining both temporal and feature-wise attention to capture both temporal dependencies and feature interactions in history.

**Self-supervised learning of time series.**  Being widely studied first for computer vision tasks (He et al., 2020; Chen et al., 2020; Grill et al., 2020; Chen* et al., 2021), self-supervised learning achieves strong performance with the advantage of not relying on labeled data. Recent works (Yue et al., 2022; Tonekaboni et al., 2021; Woo et al., 2022; Zhang et al., 2022a) further generalize and adapt self-supervised learning methods for time series, including classification, forecasting, and anomaly detection tasks. However, existing works of counterfactual outcome estimation over time have neglected self-supervised learning of time series as an effective way of learning informative representations. Meanwhile, existing models for self-supervised learning of time series are not tailored for counterfactual outcome estimation. Hence we propose SCOT to mitigate the gap.

## B  IDENTIFIABILITY ASSUMPTIONS

**Assumption B.1** (Consistency). *The potential outcome of any treatment* $\boldsymbol{a}_t$ *is always the same as the factual outcome when a subject is given the treatment* $\boldsymbol{a}_t$: $\boldsymbol{y}_{t+1}[\boldsymbol{a}_t] = \boldsymbol{y}_{t+1}$.

**Assumption B.2** (Positivity). *If* $P(\bar{\boldsymbol{A}}_{t-1} = \bar{\boldsymbol{a}}_{t-1}, \bar{\boldsymbol{X}}_t = \bar{\boldsymbol{x}}_t) \neq 0$, *then* $P(\boldsymbol{A}_t = \boldsymbol{a}_t|\bar{\boldsymbol{A}}_{t-1} = \bar{\boldsymbol{a}}_{t-1}, \bar{\boldsymbol{X}}_t = \bar{\boldsymbol{x}}_t) > 0$ *for any* $\bar{\boldsymbol{a}}_t$.

**Assumption B.3** (Sequential strong ignorability). $\boldsymbol{Y}_{t+1}[\boldsymbol{a}_t] \perp\!\!\!\perp \boldsymbol{A}_t|\bar{\boldsymbol{A}}_{t-1}, \bar{\boldsymbol{X}}_t, \forall \boldsymbol{a}_t, t$.

## C    WHY USE TEMPORALLY CAUSAL ATTENTION IN SCOT?

The embeddings of time-varying features $\boldsymbol{Z}^{S,(L)} \in \mathbb{R}^{t \times (d_X + d_A + d_Y) \times d_{\text{model}}}$ from the final layer is re-organized to stepwise representations $(\boldsymbol{Z}_1, \boldsymbol{Z}_2, \ldots, \boldsymbol{Z}_t)$. Each $\boldsymbol{Z}_{t'} = \{\boldsymbol{z}_{t'}^i \in \mathbb{R}^{d_{\text{model}}}\}_{i=1}^{d_X + d_A + d_Y}$ is further aggregated to $\boldsymbol{z}_{t'}^X, \boldsymbol{z}_{t'}^A, \boldsymbol{z}_{t'}^Y, \boldsymbol{z}_{t'}$ in Equation 1. When the encoder satisfies the temporal causality (i.e. $\boldsymbol{Z}_{t'}$ only depends on $\bar{\boldsymbol{H}}_{t'}$), they can be seen as a sequence of representations for the observed history $\bar{\boldsymbol{H}}_1, \bar{\boldsymbol{H}}_2, \ldots, \bar{\boldsymbol{H}}_t$ truncated at each time step.

When we feed the encoder with factual data in training stages, a major advantage of encoders satisfying temporal causality is that we can estimate the outcomes and evaluate the factual estimation losses in every time step of the input sequence at a single forward pass. Evaluating of counterfactual data, the encoder only needs to keep $\boldsymbol{Z}_t$ representing the entire observed history, conditioning on which the predictor rolls out outcome estimations given counterfactual treatments.

In contrast, feeding the entire history in one pass for training is error-prone for architectures and can violate the temporal causality (e.g. transformers with fully temporal attention or frequency-based methods), since it leaks information of future steps into the representations in previous steps. Predictors trained with such representations converge quickly to a trivial model that simply copies future steps as estimations. When we evaluate counterfactual data where future counterfactual outcomes are no longer available in input, the performance degenerates. As a result, we have to explicitly unroll the observed sequence to $t$ truncated sequences and run $t$ forward passes to get the representations and factual errors in every step when training non-temporally-causal models. This leads to a $\times T$ increase in training time when the batch size remains unchanged due to hardware restrictions, where $T$ is the maximum length of sequences in training data. We have $T \geq 50$ in our experiments and find that none of the architecture violating temporal causality can finish training in a reasonable time.

## D    PROOF OF GENERALIZATION BOUND OF CONTRASTIVE LEARNING IN COUNTERFACTUAL OUTCOME ESTIMATION

### D.1    UNSUPERVISED DOMAIN ADAPTATION VIEW OF COUNTERFACTUAL OUTCOME ESTIMATION

For the given observed history $\bar{\boldsymbol{H}}_t$, treatment sequence $\bar{\boldsymbol{a}}_{t:t+\tau-1}$ to apply, and the outcome $\boldsymbol{y}_{t+\tau}[\boldsymbol{a}_{t:t+\tau-1}](\bar{\boldsymbol{H}}_t)$ to estimate, we slightly abuse scalar/vector/matrix notations and denote them as $h, a, y[a](h)$ for simplicity. With discrete treatments[3], we notice that the counterfactual outcome prediction of each type of treatments from factual data can be viewed as an unsupervised domain adaptation (**UDA**) problem:

For a treatment type $a$, we aim at finding a function $f_a(h)$ that specifically estimates $\mathbb{E}(y[a](h))$. Any sample $(h_i, a_i, y_i)$ from the observed dataset $\mathcal{D}_{tr}$ can be categorized into one of (1) **Labeled subset** $\mathcal{S}_a = \{(h_i, a_i, y_i)\}_{a_i = a}$ and (2) **Unlabeled subset** $\mathcal{T}_a = \{(h_i, a_i, y_i)\}_{a_i \neq a}$. According to Assumption B.1, for any $(h_i, a_i, y_i) \in \mathcal{S}_a$, we have $y_i = y[a](h_i)$ and thus $y_i$ is a label of $f_a(h)$. In contrast, for any $(h_i, a_i, y_i) \in \mathcal{T}_a$, $y_i = y[a_i](h_i)$ and $a_i \neq a$, resulting in that $y_i$ is no longer a valid label of $f_a(h)$. For simplicity, we omit the treatment symbol $a$ as well as the $y_i$ in $\mathcal{T}_a$: $\mathcal{S}_a = \{(h_i, y_i)\}_{a_i = a}, \mathcal{T}_a = \{h_i\}_{a_i \neq a}$.

Considering the existence of treatment bias, there exists at least a $a' \neq a$ satisfying $P(h|a) \neq P(h|a')$, which potentially leads to $P_{\mathcal{T}_a}(h) = P_{\mathcal{D}_{tr}}(h_i|a_i \neq a) \neq P_{\mathcal{D}_{tr}}(h_i|a_i = a) = P_{\mathcal{S}_a}(h)$. In counterfactual estimation, we aim at minimizing the estimation error without treatment bias:

$$\mathcal{E}_{\mathcal{D}}(f_a) = \mathbb{E}_{h \sim P_{\mathcal{D}}(h), y \sim P_{\mathcal{D}}(y[a]|h)} \ell(f_a(h), y). \tag{16}$$

Here, we focus on the case where no covariate and concept shifts happen across datasets [4]: $P_{\mathcal{D}}(h) = P_{\mathcal{D}_{tr}}(h) = \sum_{a'} P(a') P_{\mathcal{D}_{tr}}(h|a'), P_{\mathcal{D}}(y[a]|h) = P_{\mathcal{D}_{tr}}(y[a]|h) = P_{\mathcal{S}_a}(y[a]|h) = P_{\mathcal{T}_a}(y[a]|h).$

---

[3]For a sequence of discrete treatments, we can always map it to a single discrete variable with a proper encoding.

[4]The distribution shift between $P_{\mathcal{D}}(h)$ v.s. $P_{\mathcal{D}_{tr}}(h)$ (covariate shift) and $P_{\mathcal{D}}(y[a]|h)$ v.s. $P_{\mathcal{D}_{tr}}(y[a]|h)$ (concept shift) can be viewed as the general covariate shift/concept shift and fit into existing theories of domain adaptation/generalization (Farahani et al., 2021).

Then Eq. 16 becomes:

$$
\begin{aligned}
\mathcal{E}_{\mathcal{D}}(f_a) &= \sum_{a'} P(a') \mathbb{E}_{h \sim P_{\mathcal{D}_{tr}}(h|a'), y \sim P_{\mathcal{D}_{tr}}(y[a]|h)} \ell(f_a(h), y) \\
&= P(a) \underbrace{\mathbb{E}_{h \sim P_{\mathcal{S}_a}(h), y \sim P_{\mathcal{D}_{tr}}(y[a]|h)} \ell(f_a(h), y)}_{\mathcal{E}_{\mathcal{S}_a}(f_a)} + \\
&\quad (1 - P(a)) \underbrace{\mathbb{E}_{h \sim P_{\mathcal{T}_a}(h), y \sim P_{\mathcal{D}_{tr}}(y[a]|h)} \ell(f_a(h), y)}_{\mathcal{E}_{\mathcal{T}_a}(f_a)}.
\end{aligned}
\tag{17}
$$

So far we can see that for the task of finding an outcome estimator for treatment type $a$, the counterfactual estimation error is bounded by estimation errors on both $\mathcal{S}_a$ (denoted as $\mathcal{E}_{\mathcal{S}_a}(f_a)$) and $\mathcal{T}_a$ (denoted as $\mathcal{E}_{\mathcal{T}_a}(f_a)$). Per our analysis above, $\mathcal{S}_a$ and $\mathcal{T}_a$ corresponds to the labeled source domain data and the unlabeled target domain data in UDA problems, where potential distribution shifts exist between $\mathcal{S}_a$ and $\mathcal{T}_a$ due to treatment bias. While $\mathcal{E}_{\mathcal{S}_a}(f_a)$ can be optimized with supervised learning using factual data in $\mathcal{S}_a$, we cannot directly optimize $\mathcal{E}_{\mathcal{T}_a}(f_a)$ with labeled data directly.

Recent works (Thota & Leontidis, 2021; Sagawa et al., 2021; Park et al., 2020; Wang et al., 2021) show that contrastive learning, as an effective self-supervised representation learning method, demonstrates strong transferability in UDA and leads to simple state-of-the-art algorithms. Considering the close connection between counterfactual outcome estimation and UDA, we develop our model SCOT based on contrastive learning and our analysis of the counterfactual estimation error bound from the recent work (HaoChen et al., 2022), where the authors provide theoretical analysis of the transferability of contrastive learning in UDA.

## D.2 PRELIMINARIES

**Positive pairs.** Pairs of semantically related/similar data samples are positive pairs in contrastive learning. In contrastive learning, positive pairs are commonly generated by applying randomized transformation on the same input (He et al., 2020; Woo et al., 2022).

For exposition simplicity, we assume the set of factual observed history $\mathcal{H}$ is a finite but large dataset of size $N$. We use $P_+$ to denote the distribution of positive pairs. $P_+$ satisfies $P_+(h, h') = P_+(h', h)$, $\forall h, h' \in \mathcal{H}$. $P_{\mathcal{H}}$ denotes the marginal distribution of $P_+$: $P_{\mathcal{H}}(h) = \sum_{h' \in \mathcal{H}} P_+(h, h')$.

**Positive-pair graph.** Following the definition in (HaoChen et al., 2022), we introduce the *postive-pair graph* as a weighted undirected graph $G(\mathcal{H}, w)$ with the vertex set $\mathcal{H}$ and the edge weight $w(h, h') = P_+(h, h')$. $w(h) = P_{\mathcal{H}}(h) = \sum_{h' \in \mathcal{H}} w(h, h')$. For any vertex subset $A$, $w(A) = \sum_{h \in A} w(h)$. For any vertex subsets $A, B$, $w(A, B) = \sum_{h \in A, h' \in B} w(h, h')$. For any vertex $h$ and vetex subset $B$, $w(h, B) = w(\{h\}, B)$.

**Generalized spectral contrastive loss.** Let $r : \mathcal{H} \to \mathbb{R}^k$ be a mapping from the input data to $k$-dimensional features. For the convenience of proof, we consider the (generalized) spectral contrastive loss proposed in (HaoChen et al., 2022):

$$
\mathcal{L}_{\sigma}(r) = \mathbb{E}_{(h, h^+) \sim P_+} \left[ \left\| r(h) - r(h^+) \right\|_2^2 \right] + \sigma \cdot R(r),
\tag{18}
$$

where the regularizer is defined as $R(r) = \left\| \mathbb{E}_{h \in P_{\mathcal{H}}}[r(h)r(h)^T] - I_{k \times k} \right\|_F^2$ and $I_{k \times k}$ is the $k$-dimensional identity matrix. Notice that the InfoNCE loss is more commonly used in empirical study (He et al., 2020; Chen et al., 2020; Chen* et al., 2021) instead of the spectral contrastive loss, and their equivalence is still an open problem with some preliminary results (Tan et al., 2023).

## D.3 DEFINITIONS AND ASSUMPTIONS

We reiterate the following definitions and assumptions in (HaoChen et al., 2022) for self-containment:

**Definition D.1** (Expansion). *Let $A, B$ be two disjoint subsets of $\mathcal{H}$. We denote the expansion, max-expansion and min-expansion from $A$ to $B$ as follows:*

$$\phi(A, B) = \frac{w(A, B)}{w(A)}, \quad \bar{\phi}(A, B) = \max_{h \in A} \frac{w(h, B)}{w(h)}, \quad \underline{\phi}(A, B) = \min_{h \in A} \frac{w(h, B)}{w(h)}. \tag{19}$$

**Assumption D.2** (Cross-cluster connections). *For some $\alpha \in (0, 1)$, we assume that vertices of the positive-pair graph $G(\mathcal{H}, w)$ can be partitioned into $m$ disjoint clusters $C_1, \ldots, C_m$ such that for any $i \in [m]$,*

$$\bar{\phi}(C_i, \mathcal{H} \backslash C_i) \leq \alpha. \tag{20}$$

**Assumption D.3** (Intra-cluster conductance). *For all $i \in [m]$, assume the conductance of the subgraph restricted to $C_i$ is large, i.e., every subset $A$ of $C_i$ with at most half the size of $C_i$ expands to the rest:*

$$\forall\, A \subset C_i \text{ satisfying } w(A) \leq w(C_i)/2, \ \phi(A, C_i \backslash A) \geq \gamma. \tag{21}$$

**Assumption D.4** (Relative expansion). *Let $S$ and $T$ be two disjoint subsets of $\mathcal{H}$, each is formed by $r$ clusters among $C_1, C_2, \ldots, C_m$ for $r \leq m/2$. Let $\rho = \min_{i \in [r]} \underline{\phi}(T_i, S_i)$ be the minimum min-expansions from $T_i$ to $S_i$. For some sufficiently large universal constant $c$, we assume that $\rho \geq c \cdot \alpha^2$ and that*

$$\rho = \min_{i \in [r]} \underline{\phi}(T_i, S_i) \geq c \cdot \max_{i \neq j} \bar{\phi}(T_i, S_j). \tag{22}$$

## D.4 Proof of Theorem 3.1

We adapt the preconditioned featurer averaging classifier in (HaoChen et al., 2022) for regression in our proof:

---

**Algorithm 1** Preconditioned feature averaging (PFA).

---

**Require:** Pretrained representation extractor $r$, unlabeled data $P_\mathcal{H}$, source domain labeled data $P_S$, target domain test data $\tilde{h}$, integer $t \in \mathbb{Z}^+$, outcome discretization granularity $\epsilon$.

1: Compute the preconditioner matrix $\Sigma = \mathbb{E}_{h \in P_\mathcal{H}}[r(h)r(h)^T]$.
2: **for** every outcome value $y_i$ corresponding to the cluster $C_i, i \in [r]$ **do**
3:      Compute the mean feature of outcome $y_i$: $b_i = \mathbb{E}_{(h,y) \sim P_S}[\mathbb{1}[\|y - y_i\|_2 \leq \epsilon] \cdot r(h)]$.
4: **end for**
5: **return** prediction $y_{i^*}$, $i^* = \arg\max_{i \in [r]} \langle r(h), \sum^{t-1} b_i \rangle$.

---

For any PFA regressor $f$ constructed with Alg. 1, we can transform it to a corresponding classifier by defining its 0-1 classification error on the target domain $T$ as:

$$\mathcal{E}_T^{01}(f) = \mathbb{E}_{(h,y) \sim P_T}[\mathbb{1}[\|y - f(h)\|_2 > \epsilon]]. \tag{23}$$

We can directly apply the main result in (HaoChen et al., 2022) and get an upper bound of the 0-1 error on the target domain:

**Theorem D.5** (Upper bound of 0-1 error on the target domain (HaoChen et al., 2022)). *Suppose that Assumption D.2, Assumption D.3, and Assumption D.4 holds for the set of observed history $\mathcal{H}$ and its positive-pair graph $G(\mathcal{H}, w)$, and the representation dimension $k \geq 2m$. Let $r$ be a minimizer of the generalized spectral contrastive loss and the regression head $f$ be constructed in Alg. 1. We have*

$$\mathcal{E}_T^{01}(f) \lesssim \frac{r}{\alpha^2 \gamma^4} \cdot \exp(-\Omega(\frac{\rho \gamma^2}{\alpha^2})). \tag{24}$$

**Lemma D.6** (Relation between the L2 regression error and 0-1 classification error). *Suppose that both $\|f(h)\|_2 \leq B$ and $\|y\|_2 \leq B$, $\epsilon < 2B$. The L2 regression error $\mathcal{E}_T(f)$ of the PFA regressor on the target domain $T$ is bounded by $\mathcal{E}_T^{01}(f)$ as:*

$$\mathcal{E}_T(f) \leq \epsilon^2 + (4B^2 - \epsilon^2)\mathcal{E}_T^{01}(f). \tag{25}$$

*Proof.*

$$
\begin{aligned}
\mathcal{E}_T(f) =& \mathbb{E}_{(h,y)\in P_T} \|y - f(h)\|_2^2 \\
\leq& \sum_{(h,y)\in T} P(h,y) \left[ \mathbb{1}[\|y - f(h)\|_2 > \epsilon] \|y - f(h)\|_2^2 \right] \\
& + \sum_{(h,y)\in T} P(h,y)(1 - \mathbb{1}[\|y - f(h)\|_2 > \epsilon])\epsilon^2 \\
\leq& \sum_{(h,y)\in T} P(h,y) \left[ \mathbb{1}[\|y - f(h)\|_2 > \epsilon]4B^2 \right] \\
& + \sum_{(h,y)\in T} P(h,y)(1 - \mathbb{1}[\|y - f(h)\|_2 > \epsilon])\epsilon^2 \\
=& \epsilon^2 + (4B^2 - \epsilon^2)\mathbb{E}_{(h,y)\in P_T} \mathbb{1}[\|y - f(h)\|_2 > \epsilon] \\
=& \epsilon^2 + (4B^2 - \epsilon^2)\mathcal{E}_T^{01}(f).
\end{aligned}
$$

□

Lemma D.6 connects the L2 error and the 0-1 error. Combining Eq. 17, Eq. 24, Eq. 25, we immediately get Theorem 3.1.

# E    DATASET DESCRIPTION

Table 4: Statistics of datasets.

| Dataset | Domain | Property | Seq Length | Train/Validation/Test Seq Num |
|---|---|---|---|---|
| Tumor growth | source | $\gamma = 10$ | 60 | 10000/1000/1000 |
| | target | $\gamma = 0$ | 60 | 100/1000/1000 |
| Semi-synthetic MIMIC-III | source | age in [20,45] | 99 | 3704/926/926 |
| | target | age≥85 | 99 | 138/347/1737 |
| M5 | source | food items | 50 | 39606/7048/7048 |
| | target | household items | 50 | 3623/3512/18005 |

We summarize the statistics and the way of introducing feature distribution shifts in Table 4.

**Tumor growth.**    We refer readers to (Bica et al., 2020; Melnychuk et al., 2022) for the complete descriptions of the pharmacokinetic-pharmacodynamic (PK-PD) model. Here we focus on how we introduce distribution shifts by adjusting the treatment bias coefficient $\gamma$.

The volume of tumor after $t$ days of diagnosis is:

$$
V(t+1) = (1 + \rho\log(\frac{K}{V(t)}) - \beta_c C(t) - (\alpha_r d(t) + \beta_r d(t)^2) + e_t)V(t), \tag{26}
$$

where $K, \rho, \beta_c, \alpha_r, \beta_r$ are parameters sampled from the prior distributions defined in (Geng et al., 2017). $e_t \sim \mathcal{N}(0, 0.01^2)$ is the noise term.

PK-PD model constructs time-varying confounding by connecting the probability of assigning chemotherapy and radiotherapy with the outcome - tumor diameter:

$$
p_c(t) = \sigma(\frac{\gamma_c}{D_{\max}}(\bar{D}(t) - \delta_c)), \quad p_r(t) = \sigma(\frac{\gamma_r}{D_{\max}}(\bar{D}(t) - \delta_r)). \tag{27}
$$

$\bar{D}(t)$ is the mean tumor diameter in the past 15 days and $D_{\max} = 13$. $\sigma$ is the sigmoid function. $\delta_c = \delta_r = D_{\max}/2$. $\gamma_c$ and $\gamma_r$ controls the importance of tumor diameter history on treatment assignment, thus control the strength of time-dependent confounding.

In Tumor growth dataset, we set $\gamma_c = \gamma_r = \gamma = 10$ to generate data in the source domain, and $\gamma_c = \gamma_r = \gamma = 0$ for the target domain. As a result, both treatment bias and the data distribution of history differs between source and target domains.

**Semi-synthetic MIMIC-III.** We split the semi-synthetic MIMIC-III dataset introduced in (Melnychuk et al., 2022) by ages of patients to the source/target domain. More specifically, we generate simulation data from patients with ages falling in $[20, 45]$ as the source domain data and simulation based on patients with ages over 85 as the target domain data. Missing values in MIMIC-III dataset is imputed with the so-called "Simple Imputation" described in Wang et al. (2020). Missing values are first forward filled and then set to individual-specific mean if there are no previous values. If the variable is always missing for a patient, we set it to the global mean.

**M5.** We adapt the M5 forecasting dataset (https://www.kaggle.com/competitions/m5-forecasting-accuracy) for treatment effect estimation over time. In M5, we select the item pricing as treatment, its sales as outcome and all other features as covariates. We aggregate the item sales by week to reduce the sequence length to the same level as the other two datasets for the convenience of evaluation. We also discretize the continuous pricing by mapping $(p_{t,i} - p_{0,i})/p_{0,i}$ to buckets divided by its 20-quantiles, where $p_{t,i}, p_{0,i}$ are the prices of item $i$ at time $t$ and at its initial sale.

To introduce the feature distribution shift, we select 5000 items in the food category as the source domain data and another 5000 items in the household category as the target domain data.

## F   BASELINES

**Baseline implementation.** We reuse the implementation in (Melnychuk et al., 2022) for evaluating all the baselines, including: MSM (Robins et al., 2000), RMSN (Lim, 2018), CRN (Bica et al., 2020), G-Net (Li et al., 2021), and Causal Transformer (CT) (Melnychuk et al., 2022).

**Hyperparameter tuning.** For all baselines, we follow the ranges of hyperparameter tuning in (Melnychuk et al., 2022) and select the hyperparameters with the lowest factual outcome estimation error on the validation set from the source domain. For each method and each dataset, the same set of hyperparameters are used in the zero-shot transfer/data-efficient transfer/standard supervised learning settings. The detailed hyperparameters used for baselines and SCOT are listed in the configuration files in our code repository. Here we list the main hyperparameters for reference.

**MSM**. There is no tuneable hyperparameter in MSM.

**RMSN**. We list the hyperparameters of RMSN in Table 5.

**CRN(ERM)**. See Table 6.

**CRN**. See Table 7.

**CT(ERM)**. See Table 8.

**CT**. See Table 9.

**G-Net**. See Table 10.

**SCOT**. See Table 11.

**Comparison of numbers of model parameters.** Here we list the number of trainable parameters in each baseline as well as SCOT in the experiments of each dataset.

## G   RESULTS OF SUPERVISED LEARNING SETUP

Table 13 shows the performance in standard supervised learning setting, with both train and test data from the source domain. Overall, SCOT outperforms other baselines in tumor growth and semi-synthetic MIMIC-III datasets. With M5, SCOT also shows comparable performance to the CT(ERM) with a 1.3% relative difference.

Table 5: RMSN hyperparameters.

|  |  | Tumor growth | Semi-synthetic MIMIC-III | M5 |
|---|---|---|---|---|
| Propensity Treatment | RNN Hidden Units | 8 | 6 | 44 |
|  | Dropout | 0.5 | 0.1 | 0.4 |
|  | Layer Num | 1 | 2 | 1 |
|  | Max Gradient Norm | 1.0 | 0.5 | 2.0 |
|  | Batch Size | 128 | 256 | 128 |
|  | Learning Rate | 0.01 | 0.01 | 0.001 |
| Propensity History | RNN Hidden Units | 24 | 74 | 92 |
|  | Dropout | 0.1 | 0.5 | 0.5 |
|  | Layer Num | 1 | 2 | 2 |
|  | Max Gradient Norm | 2.0 | 1.0 | 0.5 |
|  | Batch Size | 128 | 64 | 128 |
|  | Learning Rate | 0.01 | 0.001 | 0.01 |
| Encoder | RNN Hidden Units | 24 | 74 | 46 |
|  | Dropout | 0.1 | 0.1 | 0.1 |
|  | Layer Num | 1 | 1 | 2 |
|  | Max Gradient Norm | 0.5 | 0.5 | 0.5 |
|  | Batch Size | 64 | 1024 | 128 |
|  | Learning Rate | 0.01 | 0.001 | 0.0001 |
| Decoder | RNN Hidden Units | 48 | 196 | 45 |
|  | Dropout | 0.1 | 0.1 | 0.1 |
|  | Layer Num | 1 | 1 | 1 |
|  | Max Gradient Norm | 0.5 | 0.5 | 4.0 |
|  | Batch Size | 256 | 1024 | 256 |
|  | Learning Rate | 0.0001 | 0.0001 | 0.0001 |

Table 6: CRN(ERM) hyperparameters.

|  |  | Tumor growth | Semi-synthetic MIMIC-III | M5 |
|---|---|---|---|---|
| Encoder | RNN Hidden Units | 24 | 74 | 46 |
|  | Balancing Representation Size | 18 | 74 | 46 |
|  | FC Hidden Units | 18 | 37 | 46 |
|  | Layer Num | 1 | 1 | 2 |
|  | Dropout | 0.1 | 0.1 | 0.1 |
|  | Batch Size | 256 | 64 | 128 |
|  | Learning Rate | 0.01 | 0.001 | 0.001 |
| Decoder | RNN Hidden Units | 18 | 74 | 46 |
|  | Balancing Representation Size | 6 | 98 | 90 |
|  | FC Hidden Units | 6 | 98 | 22 |
|  | Layer Num | 1 | 2 | 1 |
|  | Dropout | 0.1 | 0.1 | 0.1 |
|  | Batch Size | 256 | 256 | 256 |
|  | Learning Rate | 0.001 | 0.0001 | 0.0001 |

Table 7: CRN hyperparameters.

| | | Tumor growth | Semi-synthetic MIMIC-III | M5 |
|---|---|---|---|---|
| Encoder | RNN Hidden Units | 18 | 74 | 46 |
| | Balancing Representation Size | 3 | 74 | 46 |
| | FC Hidden Units | 12 | 37 | 46 |
| | Layer Num | 1 | 1 | 2 |
| | Dropout | 0.2 | 0.1 | 0.1 |
| | Batch Size | 256 | 64 | 128 |
| | Learning Rate | 0.001 | 0.001 | 0.001 |
| Decoder | RNN Hidden Units | 3 | 74 | 46 |
| | Balancing Representation Size | 3 | 98 | 90 |
| | FC Hidden Units | 3 | 98 | 22 |
| | Layer Num | 1 | 2 | 1 |
| | Dropout | 0.2 | 0.1 | 0.1 |
| | Batch Size | 256 | 256 | 256 |
| | Learning Rate | 0.001 | 0.0001 | 0.0001 |

Table 8: CT(ERM) hyperparameters.

| | Tumor growth | Semi-synthetic MIMIC-III | M5 |
|---|---|---|---|
| Transformer Hidden Units | 12 | 24 | 24 |
| Balancing Representation Size | 2 | 88 | 94 |
| FC Hidden Units | 12 | 44 | 47 |
| Layer Num | 1 | 1 | 2 |
| Head Num | 2 | 3 | 2 |
| Max Relative Position | 15 | 20 | 30 |
| Dropout | 0.1 | 0.1 | 0.1 |
| Batch Size | 64 | 64 | 64 |
| Learning Rate | 0.001 | 0.01 | 0.001 |

Table 9: CT hyperparameters.

| | Tumor growth | Semi-synthetic MIMIC-III | M5 |
|---|---|---|---|
| Transformer Hidden Units | 16 | 24 | 24 |
| Balancing Representation Size | 16 | 88 | 94 |
| FC Hidden Units | 16 | 44 | 47 |
| Layer Num | 1 | 1 | 2 |
| Head Num | 2 | 3 | 2 |
| Max Relative Position | 15 | 20 | 30 |
| Dropout | 0.2 | 0.1 | 0.1 |
| Batch Size | 64 | 64 | 64 |
| Learning Rate | 0.001 | 0.01 | 0.001 |

Table 10: G-Net hyperparameters.

| | Tumor growth | Semi-synthetic MIMIC-III | M5 |
|---|---|---|---|
| RNN Hidden Units | 24 | 148 | 144 |
| FC Hidden Units | 48 | 74 | 72 |
| Dropout | 0.1 | 0.1 | 0.1 |
| Layer Num | 1 | 1 | 2 |
| Batch Size | 128 | 256 | 256 |
| Learning Rate | 0.001 | 0.01 | 0.001 |

Table 11: SCOT hyperparameters.

|  |  | Tumor growth | Semi-synthetic MIMIC-III | M5 |
|---|---|---|---|---|
| Encoder | Transformer Hidden Units | 24 | 36 | 36 |
|  | Encoder Momentum | 0.99 | 0.99 | 0.99 |
|  | Temperature | 1.0 | 1.0 | 1.0 |
|  | Layer Num | 1 | 1 | 2 |
|  | Head Num | 2 | 3 | 2 |
|  | Dropout | 0.1 | 0.1 | 0.1 |
|  | Batch Size | 64 | 64 | 64 |
|  | Learning Rate | 0.001 | 0.001 | 0.001 |
| Decoder | Hidden Units | 128 | 128 | 128 |
|  | Batch Size | 32 | 32 | 32 |
|  | Learning Rate | 0.001 | 0.001 | 0.001 |

Table 12: Number of trainable parameters.

| #trainable params | Tumor growth | semi-synthetic MIMIC-III | M5 |
|---|---|---|---|
| MSM | <1K | (-) | (-) |
| RMSN | 18.8K | 387K | 213K |
| CRN(ERM) | 6.5K | 164K | 78K |
| CRN | 2.3K | 164K | 78K |
| CT(ERM) | 5.2K | 45K | 80.3K |
| CT | 9.4K | 45K | 80.3K |
| G-Net | 3.4K | 151K | 323K |
| SCOT | 20.7K | 43.6K | 77.5K |

Table 13: Results in standard supervised learning setting, with source and target datasets coming from the same distribution for multi-step outcome estimation. We report the mean +- standard deviation of Rooted Mean Squared Errors (RMSEs) over 5 runs. **Bold**: the best results. Underline: the 2nd best results.

| Dataset | Method | $\tau=1$ | $\tau=2$ | $\tau=3$ | $\tau=4$ | $\tau=5$ | $\tau=6$ | Avg | Gain(%) |
|---|---|---|---|---|---|---|---|---|---|
| Tumor growth | MSM | $5.8368_{\pm0.6157}$ | $\mathbf{2.0400_{\pm0.6719}}$ | $\mathbf{3.0385_{\pm0.9990}}$ | $3.8701_{\pm1.2736}$ | $4.6173_{\pm1.5246}$ | $5.3823_{\pm1.7839}$ | $4.1308_{\pm1.1211}$ | 12.3% |
|  | RMSN | $4.8388_{\pm0.7770}$ | $5.4447_{\pm1.9202}$ | $5.9261_{\pm2.1096}$ | $\underline{5.9817_{\pm2.1270}}$ | $5.8705_{\pm2.0544}$ | $5.5461_{\pm1.8865}$ | $5.6013_{\pm1.7727}$ | 35.4% |
|  | CRN(ERM) | $5.1601_{\pm0.5222}$ | $6.0784_{\pm2.3196}$ | $6.4721_{\pm2.4221}$ | $6.6142_{\pm2.4206}$ | $6.5648_{\pm2.3455}$ | $6.2939_{\pm2.1955}$ | $6.1972_{\pm2.0226}$ | 41.6% |
|  | CRN | $4.9781_{\pm0.3169}$ | $11.5230_{\pm9.9577}$ | $15.2778_{\pm15.9393}$ | $17.3213_{\pm19.6976}$ | $18.5698_{\pm22.0477}$ | $19.2921_{\pm23.5647}$ | $14.4937_{\pm15.2290}$ | 75.0% |
|  | CT(ERM) | $5.1286_{\pm1.3377}$ | $5.7262_{\pm2.7601}$ | $6.5085_{\pm2.9886}$ | $6.9248_{\pm3.0009}$ | $7.1971_{\pm2.9346}$ | $7.2369_{\pm2.7570}$ | $6.4537_{\pm2.5904}$ | 43.9% |
|  | CT | $6.5485_{\pm1.5221}$ | $7.5382_{\pm2.8528}$ | $7.9030_{\pm2.9569}$ | $7.9828_{\pm2.9332}$ | $7.8244_{\pm2.8075}$ | $7.4418_{\pm2.6103}$ | $7.5398_{\pm2.5976}$ | 52.0% |
|  | G-Net | $3.9371_{\pm0.4023}$ | $3.7697_{\pm1.1861}$ | $4.6054_{\pm1.4181}$ | $4.9730_{\pm1.4773}$ | $5.0491_{\pm1.4410}$ | $\underline{4.8745_{\pm1.3153}}$ | $4.5348_{\pm1.1778}$ | 20.1% |
|  | SCOT | $\mathbf{3.7403_{\pm0.3695}}$ | $\underline{3.0067_{\pm0.9065}}$ | $\underline{3.4619_{\pm1.1557}}$ | $\mathbf{3.8501_{\pm1.3127}}$ | $\mathbf{3.9160_{\pm1.3142}}$ | $\mathbf{3.7525_{\pm1.1493}}$ | $\mathbf{3.6212_{\pm1.0040}}$ | (-) |
| Semi-synthetic MIMIC-III | RMSN | $0.2107_{\pm0.0261}$ | $0.5352_{\pm0.0842}$ | $0.6722_{\pm0.1096}$ | $0.7669_{\pm0.1203}$ | $0.8309_{\pm0.1280}$ | $0.8764_{\pm0.1331}$ | $0.6487_{\pm0.0976}$ | 21.7% |
|  | CRN(ERM) | $\mathbf{0.1951_{\pm0.0202}}$ | $0.4426_{\pm0.0799}$ | $0.5530_{\pm0.0859}$ | $\underline{0.6113_{\pm0.0842}}$ | $\underline{0.6478_{\pm0.0828}}$ | $\underline{0.6708_{\pm0.0819}}$ | $\underline{0.5201_{\pm0.0713}}$ | 2.4% |
|  | CRN | $0.3276_{\pm0.0301}$ | $0.5234_{\pm0.0839}$ | $0.6531_{\pm0.0985}$ | $0.7234_{\pm0.0985}$ | $0.7618_{\pm0.0921}$ | $0.7825_{\pm0.0854}$ | $0.6286_{\pm0.0801}$ | 19.2% |
|  | CT(ERM) | $0.2130_{\pm0.0164}$ | $0.4426_{\pm0.0766}$ | $0.5495_{\pm0.0836}$ | $0.6191_{\pm0.0851}$ | $0.6669_{\pm0.0856}$ | $0.7010_{\pm0.0834}$ | $0.5320_{\pm0.0695}$ | 4.6% |
|  | CT | $0.2175_{\pm0.0178}$ | $\underline{0.4421_{\pm0.0757}}$ | $\underline{0.5458_{\pm0.0854}}$ | $0.6161_{\pm0.0925}$ | $0.6670_{\pm0.0993}$ | $0.7047_{\pm0.1040}$ | $0.5322_{\pm0.0765}$ | 4.6% |
|  | G-Net | $0.3418_{\pm0.0290}$ | $0.6015_{\pm0.0653}$ | $0.7542_{\pm0.0758}$ | $0.8620_{\pm0.0825}$ | $0.9429_{\pm0.0875}$ | $1.0035_{\pm0.0915}$ | $0.7510_{\pm0.0686}$ | 32.4% |
|  | SCOT | $0.2286_{\pm0.0265}$ | $\mathbf{0.4417_{\pm0.0876}}$ | $\mathbf{0.5288_{\pm0.0957}}$ | $\mathbf{0.5825_{\pm0.0991}}$ | $\mathbf{0.6190_{\pm0.1018}}$ | $\mathbf{0.6458_{\pm0.1030}}$ | $\mathbf{0.5077_{\pm0.0848}}$ | (-) |
| M5 | RMSN | $35.7795_{\pm4.3603}$ | $33.2570_{\pm2.3870}$ | $33.4138_{\pm4.0678}$ | $33.4169_{\pm4.4289}$ | $33.3104_{\pm4.4017}$ | $33.3819_{\pm4.1602}$ | $33.7599_{\pm3.9379}$ | 52.2% |
|  | CRN(ERM) | $13.8445_{\pm0.1550}$ | $15.7926_{\pm0.1278}$ | $16.7071_{\pm0.2240}$ | $17.0887_{\pm0.1724}$ | $17.2709_{\pm0.0923}$ | $17.9759_{\pm0.0880}$ | $16.4466_{\pm0.1367}$ | 1.8% |
|  | CRN | $\underline{13.5907_{\pm0.0859}}$ | $15.5242_{\pm0.0692}$ | $16.2694_{\pm0.1157}$ | $\underline{16.7355_{\pm0.0719}}$ | $\underline{17.0095_{\pm0.0388}}$ | $\underline{17.6874_{\pm0.0558}}$ | $\underline{16.1361_{\pm0.0673}}$ | -0.1% |
|  | CT(ERM) | $\mathbf{13.4887_{\pm0.1335}}$ | $15.3397_{\pm0.2922}$ | $16.3415_{\pm0.4096}$ | $17.0545_{\pm0.5603}$ | $17.4828_{\pm0.5609}$ | $18.5832_{\pm0.4930}$ | $\mathbf{15.9414_{\pm0.3804}}$ | -1.3% |
|  | CT | $13.6721_{\pm0.3574}$ | $15.9384_{\pm1.0910}$ | $17.2781_{\pm1.6049}$ | $18.1796_{\pm1.8134}$ | $18.8805_{\pm2.2159}$ | $19.2510_{\pm1.9642}$ | $16.7897_{\pm1.4144}$ | 3.8% |
|  | G-Net | $13.7187_{\pm0.0833}$ | $\mathbf{14.9851_{\pm0.1205}}$ | $\mathbf{15.9578_{\pm0.1701}}$ | $16.8278_{\pm0.2229}$ | $17.4833_{\pm0.3111}$ | $18.1665_{\pm0.3795}$ | $16.1898_{\pm0.2070}$ | 0.3% |
|  | SCOT | $14.2556_{\pm0.1792}$ | $15.6151_{\pm0.2287}$ | $\underline{16.1743_{\pm0.2076}}$ | $\mathbf{16.4791_{\pm0.1276}}$ | $\mathbf{16.9037_{\pm0.1322}}$ | $\mathbf{17.4379_{\pm0.1845}}$ | $16.1443_{\pm0.1742}$ | (-) |

## H  VISUALIZATION OF THE LEARNED REPRESENTATIONS

Fig. 3 depicts the representations learned for the Semi-synthetic MIMIC-III dataset after each of the 4 stages: (a) **Pretrained**: representations after the self-supervised learning stage of source data. (b) **Non-Cold-Start**: representations fine-tuned with factual outcome estimation loss of source data. (c) **Cold-Start**: representations of target data when directly applying the encoder trained in (b). (d)

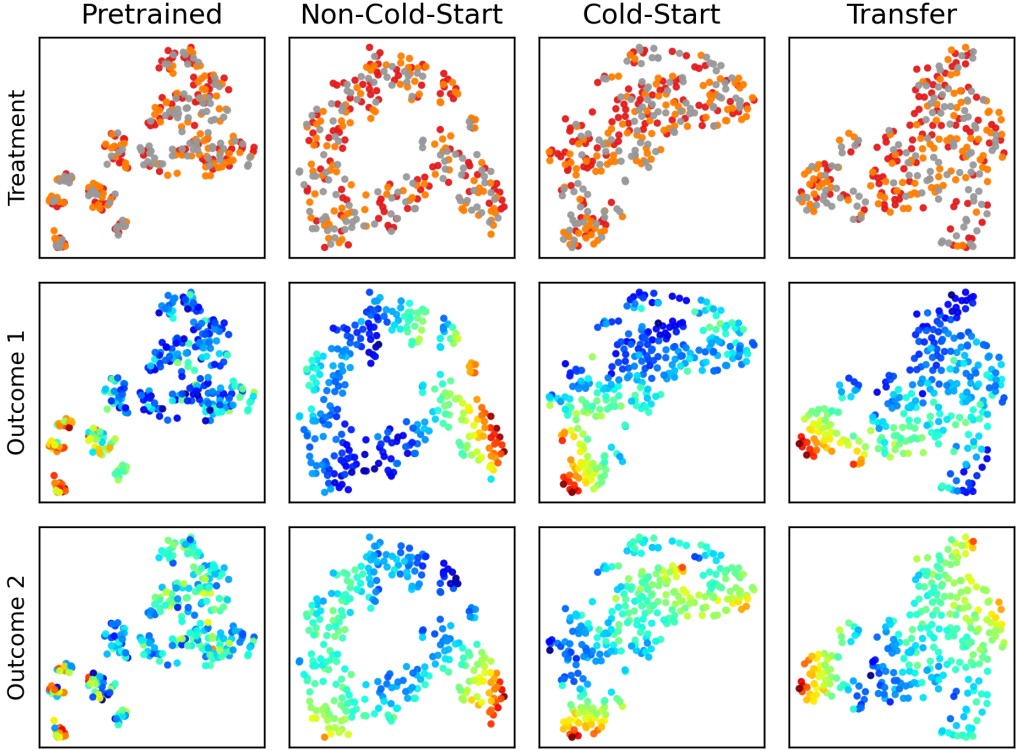

Figure 3: T-SNE visualization of learned representations in Semi-synthetic MIMIC-III dataset.

**Transfer**: representations of target data after fine-tuned with small amount of target data. We use T-SNE to map each representation to a 2D space and color each point with values of its upcoming treatment and outcomes.

As shown in the first row, representations with different types of upcoming treatments overlap, indicating that the learned representations after each stage are balanced towards treatments. In the second and the third rows, we observe clusters of representations corresponding to similar outcome values, which indicates that the learned representations are informative about the upcoming outcomes, even including the representations trained only with self-supervised loss (column "Pretrained"). Such clustered structures also persist when moving from the source domain data (column "Non-Cold-Start") to the target domain data ("Cold-Start"), showing that the learned representations can generalize to cold-start cases.

## I    EXAMPLES OF COUNTERFACTUAL TREATMENT OUTCOME ESTIMATION

Fig. 4 qualitatively compare the counterfactual outcome estimation performance differences between SCOT and baselines in the zero-shot transfer setting. We randomly select a sequence from the observed data until time $t = 4$ (x-axis), then apply sequences of treatments sampled uniformly (i.e. no treatment bias) and simulate the step-wise outcomes for 10 times as the ground truth. We compare the ground truth of each simulation with all methods tested with semi-synthetic MIMIC-III dataset. In Fig. 4 we find that the gaps between estimations and ground truth outcomes are obvious in columns of baseline results. Instead, they closely match each other in the estimation results (the rightmost column) given by SCOT, demonstrating its superior performance.

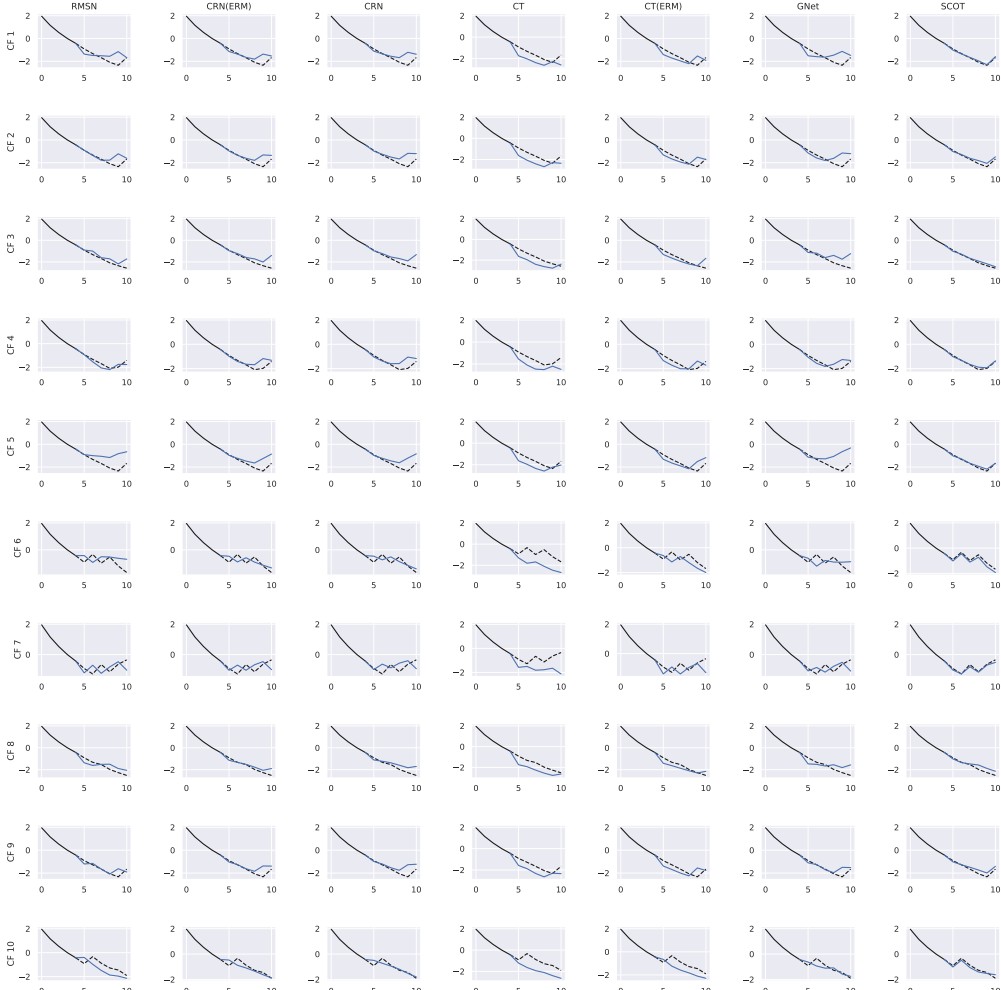

Figure 4: Examples of counterfactual treatment outcome estimation with semi-synthetic MIMIC-III data in the zero-shot transfer setting. We plot one of the two output dimensions for clarity. Each row lists the results of a counterfactual treatment sequence, while each column shows the estimations of one method across all treatment sequences tested. In each sub-figure, the observed historical outcomes are plotted in black solid lines, and the ground truth counterfactual outcomes in black dash lines. The blue solid lines show the estimated outcomes.

