# OpenReview forum: "SCOT: Improved Temporal Counterfactual Estimation with Self-Supervised Learning"
_ICLR.cc/2024/Conference — Submitted to ICLR 2024_

### Official Review · Reviewer_vBAj · 2023-10-24

**Soundness:** 3 good
**Presentation:** 3 good
**Contribution:** 3 good
**Rating:** 6
**Confidence:** 5

**Summary:**

The paper discusses the challenge of accurately estimating treatment outcomes over time based on observed historical data in various fields like medicine and e-commerce. While randomized controlled trials are the ideal method, they are often impractical. Therefore, using available data, such as electronic health records or sales history, has gained interest. Estimating treatment outcomes from time series data presents unique challenges due to complex dynamics and dependencies. Existing approaches use neural networks and different training strategies to account for time-dependent factors. However, they rely heavily on supervised learning, which limits their applicability in cases with limited or no testing data. The proposed solution, Self-supervised Counterfactual Transformer (SCOT), represents a shift from supervised to self-supervised training. SCOT uses an encoder architecture with temporal and feature-wise attention to capture dependencies in both time and features. It refines the contrastive loss in self-supervised learning, comparing the entire history and individual components of covariates, treatments, and outcomes. Additionally, the paper considers the counterfactual outcome estimation problem from an unsupervised domain adaptation perspective and provides a theoretical analysis of the error bound for such estimators based on self-supervised learning representations.

**Strengths:**

The propose methodology is following the potential outcomes framework proposed by Splawa-Neyman et al., 1990 and Rubin, 1978, extended to time-varying treatments and outcomes. Rather than directly estimating the counterfactual outcome based on a structural model, the core idea is to learn high-quality representations of observed history sequences that are informative for counterfactual treatment outcome estimation. The idea of integrating self-supervised learning (SSL) with component-specific contrastive losses is novel, which could be used to learn more informative historical representations in the temporal counterfactual outcome estimation. The paper has also provided comprehensive empirical findings using both synthetic and real-world datasets, showcasing the encouraging effectiveness of the proposed approach.

**Weaknesses:**

The overarching idea shares some similarities with approaches found in works like "Predicting Treatment Responses Over Time with Recurrent Marginal Structural Networks" and "Estimating Counterfactual Treatment Outcomes over Time Through Adversarially Balanced Representations." In these methods, the primary objective is also to discover an effective representation of historical data. However, what sets the present approach apart is its use of a transformer architecture, a notably superior choice given that transformers have demonstrated better performance than recurrent neural networks in various machine learning tasks.

**Questions:**

NA

---

> ### Author Response · Authors · 2023-11-17
> **We appreciate the reviewer for providing positive feedback on our work.**
>
> We appreciate the reviewer for providing positive feedback on our work. Here we want to restate our main contributions over existing works such as RMSN and CRN as follows: (1) As far as we know, our work is the first to adapt self-supervised learning (SSL) in temporal counterfactual outcome estimation. Instead of simply putting SSL in our task, we also provide theoretical insights for the contribution of SSL in Section 3.4. (2) In addition to SSL, we also propose a new encoder architecture combining both temporal and feature-wise attention to capture both temporal dependencies and feature interactions in the observed history. We have also added the above discussions in Related Work (Appendix A) in our revised version.

---

### Official Review · Reviewer_ZQ5p · 2023-10-30

**Soundness:** 3 good
**Presentation:** 3 good
**Contribution:** 1 poor
**Rating:** 3
**Confidence:** 5

**Summary:**

The paper proposes SCOT, a method extending the causal transformer [1] with self-supervised learning by applying random augmentations of the time series data to improve the prediction of time varying potential outcome estimation. Extensive experiments on 3 datasets show strong performance compared to existing baselines.

**Strengths:**

1. The paper tackles an important issue with potential outcomes estimation on time-varying confounded data.
2. The experimental evaluation of the method is exhaustive.
3. The method shows substantial improvements over existing baselines.

**Weaknesses:**

1. The main weakness of the paper seems to be that contribution is only incremental. Applying existing random augmentations with the component-wise contrastive loss is quite straightforward while the main part of method seems to come from the work of [1].
2. The motivation of substantial parts of the work is often not clear and could be improved. E.g., what are the exact improvements over [1] and why are these expected to work better. Also, the used terminology for describing the setting (“cold-start”, “zero-shot transfer”, “source/target domain”) is a bit confusing and its not quite clear how this differs from basic potential outcomes evaluation (e.g., in-sample vs out-of-sample evaluation). Especially wrt to the analogy to unsupervised domain adaption it is important to describe the exact setting more specifically (in Sec. 2 as well as in Sec 3.4., e.g., distinguish between the domain shift between the distributions conditioned on the treatments induced by the confounding bias vs. the domain shift stemming from the unconditional feature distribution shifts in source and target domains). Hence, source and target domains should be clearly defined in the Problem Formulation Sec. 2.
3. For reproducibility, the code for the experiments should be provided.

The paper contribution should be spelled out better. If 2. the authors claim their main contribution is around UDA, why not use a better experimental setting (and more common datasets that align with UDA-ts)? Why not review the UDA for time-series stream in the related work? Why build on SOTA methods for UDA-ts? If the authors say that their main contribution is around self training ,why not use a proper method that is more grounded in the causal graph (given that the methods for data augmentation are largely 1:1 copies from Woo et al)?

[1] Melnychuk, Valentyn, Dennis Frauen, and Stefan Feuerriegel. "Causal transformer for estimating counterfactual outcomes." International Conference on Machine Learning. PMLR, 2022.

**Questions:**

1. In Figure 3, the representations trained only on the self-supervised objective seem already unpredictive of the treatment. This indicates that either the dataset does not contain a high-level of confounding by default or the self-supervised learning can already balance the representations. If the latter, possible (theoretical) intuition why this works would be interesting, especially to support the use of self-supervised learning in this setting.
2. In general, the paper could adapt more to the terminology of counterfactual vs interventional distribution of the causal ladder of Pearl. E.g., in the Figure 1, the “cold-start /UDA case” is an interventional question (what will happen if treatment A will be prescribed) and not a counterfactual question (what would have happened if instead of treatment B  treatment A had been prescribed).
3. I assume that in general, the rMSE is evaluated on unconfounded test data for the semi-synthetic datasets. In Sec 4.2., does the data in the target domain leveraged for fine-tuning still contain confounding bias s.t. that the final test dataset still has a distribution shift wrt. to confounding to this data or is the confounding bias removed, here, too?

---

> ### Author Response · Authors · 2023-11-19
> **Thank you for providing feedback and suggestions to improve the submission. Here we list our responses to your major concerns.**
>
> Thank you for providing feedback and suggestions to improve the submission. Here we list our responses to your major concerns.
>
> 1. **“Contribution is incremental. Applying existing random augmentations with the component-wise contrastive loss is quite straightforward while the main part of method seems to come from Causal Transformer”.**
>
> 2. **“What are the exact improvements over Causal Transformer and why?”**
>
> **(Responses to both 1 and 2)** Here we want to restate our main contributions as follows:
>
> (1) To the best of our knowledge, our work is the first to adapt self-supervised learning (SSL) in temporal counterfactual outcome estimation. Instead of simply putting SSL in our task, we also provide theoretical insights for the contribution of SSL in Section 3.4. We demonstrate that the counterfactual outcome estimation problem can be viewed as an unsupervised domain adaptation (UDA) problem if we regard the outcome estimation of each treatment value as a separate task. Therefore, we can adapt the generalization error of SSL in UDA settings for counterfactual estimation settings.
>
> (2) In addition to SSL, we also propose a new encoder architecture combining both temporal and feature-wise attention to capture both temporal dependencies and feature interactions in the observed history.
>
> (3) Our work is significantly different from Causal Transformer (CT) [1] in both methodology and empirical results.
> - (a) The encoder of CT only considers attention across groups of covariate/treatment/outcome variables, while our proposed method considers a finer-grained level of feature interactions via attention across each input variable.
> - (b) CT utilizes the adversarial domain confusion loss to learn a balanced representation of history to alleviate treatment bias and generalize to counterfactual cases. Adversarial training involves 2 conflicting objectives: the learned representations should be able to predict outcomes but not treatments, which improves the difficulty of training. We empirically find that such a loss cannot lead to robustly improved performance. In Table 1, the complete CT model performs slightly better than its variant trained without the adversarial loss (named CT(ERM)) only in the zero-shot transfer setup on M5 data and worse in the other settings. Different from CT, we provide a novel view of the connection between UDA and counterfactual estimation (Section 3.4) and choose SSL, which is already proven to be effective in UDA, to improve generalization to counterfactual outcome estimation. We also verify our conclusion in the ablation study. In Table 3, we verify that SSL brings performance improvement of 4.4% and 0.2% on Semi-synthetic MIMIC-III and M5 respectively.
>
> 3. **“Explain the settings of “cold-start”, “zero-shot transfer”. How this differs from basic potential outcomes evaluation (e.g., in-sample vs out-of-sample evaluation).”**
>
> Our evaluations focus on the out-of-sample data, following the common practice of previous works [1-3]. In addition to the out-of-sample setting, we further consider scenarios where distributions of history of training/test samples are different. Under this setting, source/target domains refer to the distribution of history in training/test data respectively.
>
> 4. **“Especially wrt to the analogy to unsupervised domain adaption it is important to describe the exact setting more specifically”**
>
> We have revised the draft to distinguish between the domain shift from the confounding bias and that from the unconditional feature distribution shifts.
> The source/target domains in Section 3.4 are used as analogical names to describe the labeled and unlabeled subsets regarding a treatment value in the unsupervised domain adaptation framework to help us analyze the error of counterfactual outcome estimation, and are different from the definitions we use in Section 2. In Section 2, source/target domains describe the different distributions of the observed history between train/test data (i.e. the feature distribution shifts).
> For synthetic and semi-synthetic data, we always evaluate with data after removing treatment bias, while for real-world data we can only evaluate with factual examples. In both cases, settings “zero-shot transfer” and “data-efficient transfer” add feature distribution shifts between train and test, while the “standard supervised learning” setting has no unconditional feature distribution changes (results are reported in Table 6 in Appendix G).
>
> 5. **“For reproducibility, the code for the experiments should be provided.”**
>
> We will release the code and data as soon as our work is published.

---

> > ### Author Response · Authors · 2023-11-19
> > **(Cont.)**
> >
> > 6. **“If the authors say that their main contribution is around self training ,why not use a proper method that is more grounded in the causal graph?”**
> >
> > One of our main contributions is to adapt self-supervised learning (SSL) for counterfactual outcome estimation over time. We provide theoretical insights for the contribution of SSL in Section 3.4 with the help of a UDA view of the counterfactual outcome estimation task, and we verify its contribution in ablation studies. Our results reveal that SSL-learned representations of history help alleviate the negative impact of the correlation between **history** and **future treatments**. We agree that digging into the internal causal graph of the observed history and developing data augmentation methods grounded in the internal graph would be a promising direction.
> >
> > 7. **“This indicates that either the dataset does not contain a high-level of confounding by default or the self-supervised learning can already balance the representations. If the latter, possible (theoretical) intuition why this works would be interesting, especially to support the use of self-supervised learning in this setting.”**
> >
> > The reason for using self-supervised learning (SSL) is to leverage its ability to learn generalizable representations, which is important for both (i) counterfactual outcome estimation and (ii) cold-start scenarios. For (i), we provide a novel view to connect the counterfactual outcome estimation problem to the unsupervised domain adaptation (UDA) problem in Section 3.4 and show that the generalization error bound of counterfactual outcome estimation can be derived from the error bound of target domain data in the UDA problem. For (ii), the cold-start scenario is a more direct case of UDA, and using SSL again benefits the performance with its generalizable representations.
> >
> > As we have discussed, existing works such as CRN and CT use adversarial losses to learn the balanced representations: representations should maintain the ability to predict outcomes but not treatments, which loses part of the history information and limits the outcome estimation ability, as well as increasing the training difficulties. In contrast, the SSL stage in our method enables generalization to counterfactual data without the cost of adversarial training. To evaluate the ability of treatment prediction of learned representations, we train a linear probe on representations and use the average accuracy of predicting each type of treatment as the metric. We use the same architecture of SCOT, extract the output of the encoder on the target domain data as the representations, and compare their accuracies in the following settings (i) trained with outcome prediction losses on source domain data without self-supervised learning; (ii) trained with self-supervised learning losses only; (iii) trained with self-supervised learning losses first and supervised outcome prediction losses later on source domain data (the zero-shot setting). The accuracies of treatment prediction are (i) 70.5% (ii) 71.4% and (iii) 71.1%. The results show that the self-supervised learning stage even slightly improves the treatment prediction ability of representations. This also verifies the conclusion in existing work [5] that the improved generalization of contrastive pre-training is not the result of invariant representations as conventional UDA methods.
> >
> > 8. **“In general, the paper could adapt more to the terminology of counterfactual vs interventional distribution of the causal ladder of Pearl.”**
> >
> > First, we would like to emphasize that for cold-start cases (zero-shot setting), both (1) treatment bias in training and (2) feature distribution shifts exist. With the existence of (1), estimating the outcome of all possible treatments from observed data with treatments prescribed with biases toward history is a counterfactual problem.
> >
> > In addition, as we have stated in Section 2, we follow the commonly used potential outcome framework extended to time-varying data, where we estimate the **average/expectation of treatment outcomes over a certain population conditioning on the observed history** instead of the outcomes of individuals described in the causal ladder of Pearl.
> >
> > We have also seen the terminology “counterfactual” widely used in existing works [1-3] addressing the same treatment outcome estimation over time tasks and we keep it for consistency.
> >
> > 9. **“Does the data in the target domain leveraged for fine-tuning still contain confounding bias?”**
> >
> > For test data, we always remove the confounding bias in both source and target domains if the data is synthetic/semi-synthetic. For the real-world dataset, since we have no access to the ground-truth outcomes of arbitrary treatments, we have to use factual data in evaluation and it might contain confounding bias.

---

> > > ### Author Response · Authors · 2023-11-19
> > > **References**
> > >
> > > References:
> > >
> > > [1]. Melnychuk, Valentyn, Dennis Frauen, and Stefan Feuerriegel. "Causal transformer for estimating counterfactual outcomes." International Conference on Machine Learning. PMLR, 2022.
> > >
> > > [2]. Bica, I., Alaa, A. M., Jordon, J., & van der Schaar, M. (2020). Estimating counterfactual treatment outcomes over time through adversarially balanced representations. In International Conference on Learning Representations.
> > >
> > > [3]. Lim, B. (2018). Forecasting treatment responses over time using recurrent marginal structural networks. Advances in neural information processing systems, 31.
> > >
> > > [4]. Robins, J., & Hernan, M. (2008). Estimation of the causal effects of time-varying exposures. Chapman & Hall/CRC Handbooks of Modern Statistical Methods, 553-599.
> > >
> > > [5]. Shen, K., Jones, R. M., Kumar, A., Xie, S. M., HaoChen, J. Z., Ma, T., & Liang, P. (2022, June). Connect, not collapse: Explaining contrastive learning for unsupervised domain adaptation. In International Conference on Machine Learning (pp. 19847-19878). PMLR.

---

> ### Comment · Reviewer_ZQ5p · 2023-11-21
>
> I thank the authors for their extensive answers, which made many aspects of the paper more clear to me. In principle, I think this is an interesting paper with nice experimental evaluation and reasonable motivation (even though it could be still improved at some points). Hence, I am raising my score to 5.
> However, I would like to emphasize that, while nicely validated, the contribution of the paper itself is only marginal and not sufficient for an ICLR paper for the following reasons:
>
> **1. Contribution: Architecture.**
> It is important to emphasize that the **main parts** of the proposed architecture have been originally proposed by [1]. The architectural contribution of this work here is the featurewise attention only. Here, it is important to mention that (i) featurewise attention in general has also been used for treatment effect estimation in previous works (e.g., [2]), and (ii) the gains of the feature-wise attention blocks are only marginal compared to the original temporal attention blocks from the causal transformer of [1].
>
> **2. Contribution: Self-supervised learning for treatment effect estimation and link to UDA.**
> I think that self-supervised learning for more expressive and robust expression representation learning for time-varying treatment effect estimation definitely is an useful approach. However, also here the approach lacks novelty in the proposed points: (i) the authors state to provide a novel view of the connection between UDA and estimation of potential outcomes (Sec. 3.4). But there exist already works discussing this connection (e.g., [3, 4]) in more detail. (ii) Then, the authors simply leverage already existing results for contrastive learning and unsupervised domain adaptation. Also, there exists already closely related work for self-supervised contrastive learning for unsupervised domain adaptation of time series data (e.g., [5]) benefitting from this connection. (iii) The applied data augmentations are not novel [6]. Here would be an opportunity to use customized augmentations grounded in the causal graph of the treatment effect setting. (iv) The applied component-wise contrastive loss is quite straightforward per intuition but not quite clear why here an improvement over the simple contrastive loss is expected. Also, the improvements of this component seem to be not that significant compared to the other ablations in table 3.
> (Additionally, the code should be provided for such a submission for reproducibility and comparison to the existing works.)
>
> For now, my questions are answered. We will carefully consider the above points, as well as the rebuttal / revised paper, in the discussion period!
>
> [1] Melnychuk, Valentyn, Dennis Frauen, and Stefan Feuerriegel. "Causal transformer for estimating counterfactual outcomes." International Conference on Machine Learning. PMLR, 2022.
>
> [2] Zhang, Yi-Fan, et al. "Exploring transformer backbones for heterogeneous treatment effect estimation." arXiv preprint arXiv:2202.01336 (2022).
>
> [3] Johansson, Fredrik D., et al. "Generalization bounds and representation learning for estimation of potential outcomes and causal effects." The Journal of Machine Learning Research 23.1 (2022): 7489-7538.
>
> [4] Johansson, Fredrik, Uri Shalit, and David Sontag. "Learning representations for counterfactual inference." International conference on machine learning. PMLR, 2016.
>
> [5] Ozyurt, Yilmazcan, Stefan Feuerriegel, and Ce Zhang. "Contrastive Learning for Unsupervised Domain Adaptation of Time Series." International Conference on Learning Representations. 2022.
>
> [6] Woo, Gerald, et al. "CoST: Contrastive Learning of Disentangled Seasonal-Trend Representations for Time Series Forecasting." International Conference on Learning Representations. 2022.

---

> ### Author Response · Authors · 2023-11-23
> **We greatly appreciate the reviewer for carefully evaluating our responses! We would also like to kindly request the reviewer to update the score in the original response.**
>
> We greatly appreciate the reviewer for carefully evaluating our responses and for providing comprehensive feedback. Still, here we would like to reillustrate the novelty of our work. In short, our SSL-based method is the first to address the counterfactual outcome estimation problem for time series when both treatment bias and feature distribution shifts exist.
>
> We are ready to share our code upon request once the double-blind review stage ends.
>
> We would also like to kindly request the reviewer to update the score in the original response to reflect the change in the review.
>
> > **Contributions in the model architecture.**
>
> We would like to emphasize that our proposed encoder architecture considers **both temporal and feature-wise** interactions for the **temporal counterfactual outcome estimation** task.
> - (a) The temporal attention as well as the relative positional encoding are first proposed in [7] and [8] respectively, and they are the only two parts our model shares with [1].
> - (b) Our model utilizes the temporal attention and the feature-wise attention alternatingly to capture both temporal and feature-wise interactions. Instead, [2] only considers feature-wise interactions in a static setting.
> - (c) We added the ablation study results showing the effect of temporal/feature-wise attention blocks:
>
> - Semi-Synthetic MIMIC-III:
>
> ||$\tau=1$|2|3|4|5|6|Avg|Gain(%)|
> |-|-|-|-|-|-|-|-|-|
> |SCOT |0.2266+-0.0249|0.4501+-0.0893|0.5406+-0.0987|0.5964+-0.1020|0.6344+-0.1040|0.6637+-0.1052|0.5186+-0.0869|(-)|
> |Temporal blocks only|0.2729+-0.0409|0.4711+-0.0836|0.5607+-0.0937|0.6160+-0.0995|0.6553+-0.1038|0.6831+-0.1043|0.5432+-0.0856|4.53%|
> |Feature-wise blocks only|1.1210+-0.0827|1.1287+-0.0855|1.1755+-0.1076|1.2055+-0.1251|1.2296+-0.1418|1.2547+-0.1566|1.1858+-0.1155|56.3%|
>
> - M5:
>
> ||$\tau=1$|2|3|4|5|6|Avg|Gain(%)|
> |-|-|-|-|-|-|-|-|-|
> |SCOT |6.4054+-0.0547|6.9328+-0.0634|7.2428+-0.0700|7.4585+-0.0580|7.7012+-0.0627|7.8278+-0.0651|7.2614+-0.0609|(-)|
> |Temporal blocks only|6.4085+-0.0538|6.9547+-0.0535|7.2673+-0.0453|7.4825+-0.0388|7.7167+-0.0380|7.8328+-0.0430|7.2771+-0.0409|0.2%|
> |Feature-wise blocks only|6.8805+-0.0333|7.6298+-0.0212|7.9706+-0.0254|8.1215+-0.0298|8.3989+-0.0411|8.5303+-0.0435|7.9219+-0.0311|8.3%|
>
>
> > **Contributions in SSL for counterfactual outcome estimation.**
>
> - (a) We would like to clarify that the novelty of our view connecting UDA and potential outcome estimation is that we provide a bound of the counterfactual outcome estimation error **when the feature extractor is optimized using SSL**. Theorem 3.1 shows that when the feature extractor minimizes the contrastive loss, the error of counterfactual outcome estimation is bounded by the error of outcome estimation on factual data. It justifies our steps that (1) first train the feature extractor with SSL loss and (2) then train both the feature extractor and the predictor with factual outcome estimation loss. Meanwhile, both the contrastive loss and the factual outcome estimation loss can be easily generalized to multi-valued or continuous treatments by selecting S-Learner-like model architectures.
>
> Instead, while [3,4] also discusses the connection between UDA and potential outcome estimation, they derive a bound composed of both factual outcome estimation and **explicit distributional distance regularization**, which leads to regularization-based methods like BNN[4] and CFR[3]. Moreover, both [3,4] use the assumption that the treatment is binary, and it requires extra steps to extend the distributional distance regularization term beyond binary treatments in practice [3].
>
> - (b) While there exist works addressing UDA of time series forecasting/classification with SSL, as we have discussed in related work, our work is still novel in that we are not solving the prediction problem, but the counterfactual outcome estimation problem. Here **both treatment bias and feature distribution shift** exist, and we are the first to address both issues with SSL and provide competitive empirical results. While the effect of SSL on feature distribution shift has been widely discussed in the standard UDA setting, we further provide theoretical justification explaining why SSL also works for treatment bias by taking the UDA view.

---

> > ### Author Response · Authors · 2023-11-23
> > **(Cont.) We greatly appreciate the reviewer for carefully evaluating our responses! We would also like to kindly request the reviewer to update the score in the original response.**
> >
> > **References:**
> >
> > [1] Melnychuk, Valentyn, Dennis Frauen, and Stefan Feuerriegel. "Causal transformer for estimating counterfactual outcomes." International Conference on Machine Learning. PMLR, 2022.
> >
> > [2] Zhang, Yi-Fan, et al. "Exploring transformer backbones for heterogeneous treatment effect estimation." arXiv preprint arXiv:2202.01336 (2022).
> >
> > [3] Johansson, Fredrik D., et al. "Generalization bounds and representation learning for estimation of potential outcomes and causal effects." The Journal of Machine Learning Research 23.1 (2022): 7489-7538.
> >
> > [4] Johansson, Fredrik, Uri Shalit, and David Sontag. "Learning representations for counterfactual inference." International conference on machine learning. PMLR, 2016.
> >
> > [5] Ozyurt, Yilmazcan, Stefan Feuerriegel, and Ce Zhang. "Contrastive Learning for Unsupervised Domain Adaptation of Time Series." International Conference on Learning Representations. 2022.
> >
> > [6] Woo, Gerald, et al. "CoST: Contrastive Learning of Disentangled Seasonal-Trend Representations for Time Series Forecasting." International Conference on Learning Representations. 2022.
> >
> > [7]. Ashish Vaswani, Noam Shazeer, Niki Parmar, Jakob Uszkoreit, Llion Jones, Aidan N Gomez, Łukasz Kaiser, and Illia Polosukhin. Attention is all you need. Advances in neural information processing systems, 30, 2017.
> >
> > [8]. Peter Shaw, Jakob Uszkoreit, and Ashish Vaswani. Self-attention with relative position representations. In Proceedings of the 2018 Conference of the North American Chapter of the Association for Computational Linguistics: Human Language Technologies, Volume 2 (Short Papers), pp. 464–468, 2018.

---

### Official Review · Reviewer_YjCN · 2023-11-01

**Soundness:** 3 good
**Presentation:** 3 good
**Contribution:** 3 good
**Rating:** 5
**Confidence:** 5

**Summary:**

This paper presents a contrastive learning based counterfactual estimation using time sequences. The main contributions come from temporal attention + feature-wise attention in the encoder architecture, and the component-wise contrastive learning.

**Strengths:**

This paper has several strengths:
1: The design of encoder sounds convincing- we need to consider both the feature level as well as the temporal level information.
2: The component-wise constrastive learning is innovative, however, from table 3 there is insignificant improvement from the introduced component-wise constrastive learning. This does not hurt its novelty, but makes it less significant as this is claimed as the top 1 contribution. I would suggest the authors to emphasize less on this part because it brings little improvement. Instead, if you delve into table 3, you will see that the main contributors are actually the encoder design and the supervision loss. I would rather elaborate more on these parts.
3: The discussion on the supervision loss and positional embedding is comprehensive. I appreciate the extensive experiments that authors have conducted to validate these.

**Weaknesses:**

1: Figure 1 is not intuitive. What do these symbols (cross, needle and dashed line) mean?

2: As mentioned above, I would challenge the necessity of component-wise constrastive loss because it brings little improvement. If this is one of the main contributions.

3: Some parts are unclear and can be confusing. For the outcome predictor, Ht already includes treatment sequences, what is the movitation and justification for remodeling it using 2 convolution blocks? Does it imply that the latent representation of Ht could not capture the treatment sequences well? If not, can you include this in the ablation study?

4: I think the Assumption B.1 is too strong. Correct me if I am wrong, but I believe it removes uncertainty. However, when given a treatment, there is still probabilities, being either high or low, of different outcomes. In that sense, B.1 is much stronger than the no unmeasured counfounding that people usually use.

5: Figure 3: Pretrained is worse than the rest in separation, but there is not much difference among the remaining, right?

6: I think the reporting in table 3 sounds manipulated - in each dataset, you cherrypick the best-performing supervision strategy as "SCOT", and the worse-performing variants as "ablation study". This really looks unprofessional to me. I would rather fix one as SCOT across all datasets.

**Questions:**

1: As mentioned in the strengths, I think the main contributors (from the table 3) is the encoder design, as you stack two blocks of encoders (of course you have multiple layers as well) to capture feature-wise and temporal information. This can bring more room for discussion: i) is the performance gain from model complexity (if you use much more parameters), or one of the encoders? ii) in ablation study, why don't you remove one of the encoder blocks (just stacking feature-wise block or temporal block) and see how it performs? That will be more interesting.
2: How do you deal with missingness in the time series? This is common in healthcare dataset.
3: Since we talked about pretraining and transfer learning, instead of showing t-sne plots which hardly differentiates NCS, CS and Transfer, I would rather want to know the model performance of pretraining without finetuning.
4: This is minior but in Table 3 fold 2, the best one for MIMIC is actually sq. inv.

---

> ### Author Response · Authors · 2023-11-17
> **Thank you for providing feedback and suggestions to improve the submission. Here we list our responses to your major concerns.**
>
> Thank you for providing feedback and suggestions to improve the submission. Here we list our responses to your major concerns.
>
> 1. **“What do symbols (cross, needle, and dashed line) in Figure 1 mean?”**
> In Figure 1, we use an example from the healthcare domain to illustrate the treatment outcome estimation task: given the observed history of vital signs of a patient, we want to estimate the future outcomes after giving the patient one or multiple steps of treatment. Here the symbol cross/needle refers to the value of a binary treatment (giving the medicine/doing nothing). Dashed lines refer to the results of counterfactual outcome estimation.
> 2. **“The necessity of component-wise contrastive loss.”**
> We introduce the component-wise contrastive loss as a part of the self-supervised contrastive learning method. In Section 3.4 we also provide theoretical insights that the contrastive learning benefits the model in generalizing to counterfactual outcome estimation. In Table 3 we observe that the component-wise contrastive loss contributes to a 2.6% performance gain on Semi-Synthetic MIMIC-III data and 0.7% with M5 data. Notice that Semi-Synthetic MIMIC-III allows us to evaluate the exact counterfactual estimation performance, while the M5 is from real-world and thus we can only evaluate with factual data, the former scenario weighs more on the need for generalization to counterfactual estimation and demonstrates a larger improvement. Moreover, we observe that the overall variance of results on both datasets decreases after including the component-wise contrastive loss (0.0926->0.0869, 0.0715->0.0609).
> While the improvement from component-wise contrastive loss is smaller compared to the contributions from the encoder and the loss, previous works like Causal Transformer [1] also demonstrate that the improvement from the design that helps generalize to counterfactual estimation (for Causal Transformer it is the counterfactual domain confusion loss) is also relatively small. In Table 1 of [1], the performance of the complete model (CT(ours)) and the variant without counterfactual domain confusion loss (CT(alpha=0)) is comparable in all horizons.
> 3. **“For the outcome predictor, $\bar{\mathbf{H}}_t$ already includes treatment sequences, what is the motivation and justification for remodeling it using 2 convolution blocks?”**
> We would like to clarify that $\bar{\mathbf{H}}\_t$ only includes the treatment sequences in the observed history, while the decoder takes future steps of treatment as input. In other words, Ht encodes historical treatments, while the decoder predicts based on both history and future treatments. As shown in Figure 2 (a) and (c), if we estimate outcomes of future \tau steps after time t, Ht encodes the historical sequence of treatment from time $1 \dots t-1$, while the decoder uses $(a_t, …, a_{t+\tau-1})$.
> 4. **“Assumption B.1 is too strong.”**
> We would like to clarify that Assumption B.1 (consistency) connects the potential outcome and the observed outcome, which is the fundamental assumption in causal inference to ensure identifiability. Instead of removing uncertainty, it implies that the variation within treatments would not result in a different outcome [2]. In addition, all three assumptions (B.1, B.2, B.3) are commonly used in existing works to ensure identifiability [1,3,4] in time-varying data.
> 5. **“Figure 3: Pretrained is worse than the rest in separation, but there is not much difference among the remaining, right?”**
> In terms of the separation of outcome values, we observe in Figure 3 that pretrained embeddings are less separated since the pretraining stage does not involve the supervised training with outcome labels. After tuning the pretrained embeddings with supervised outcomes, the embeddings are more predictive of outcomes and thus are better separated w.r.t outcome values.
> 6. **“Reporting in table 3 sounds manipulated.”**
> As we describe in Section 3.3, we treat the weights of stepwise losses as hyperparameters and can be selected based on the validation errors. Since we cannot enumerate all possible choices of weights, we consider three simple schemes introduced in Section 4.3 (uni/inv/sq.inv) in our experiments and select the one with the lowest validation error in each experiment.

---

> > ### Author Response · Authors · 2023-11-17
> > **(Cont.) Thank you for providing feedback and suggestions to improve the submission. Here we list our responses to your major concerns.**
> >
> > 7. **“Question 1: encoder design.”**
> > (i) In Table 5 in the appendix, we report the number of parameters of baselines and our model. We control the model complexity so that the number of parameters in our model is comparable to or even fewer than the baselines. (ii) We provide the ablation study results of only keeping the temporal/feature-wise attention blocks as follows:
> >
> > Semi-Synthetic MIMIC-III:
> >
> > ||$\tau=1$|2|3|4|5|6|Avg|Gain(%)|
> > |-|-|-|-|-|-|-|-|-|
> > |SCOT |0.2266+-0.0249|0.4501+-0.0893|0.5406+-0.0987|0.5964+-0.1020|0.6344+-0.1040|0.6637+-0.1052|0.5186+-0.0869|(-)|
> > |Temporal blocks only|0.2729+-0.0409|0.4711+-0.0836|0.5607+-0.0937|0.6160+-0.0995|0.6553+-0.1038|0.6831+-0.1043|0.5432+-0.0856|4.53%|
> > |Feature-wise blocks only|1.1210+-0.0827|1.1287+-0.0855|1.1755+-0.1076|1.2055+-0.1251|1.2296+-0.1418|1.2547+-0.1566|1.1858+-0.1155|56.3%|
> >
> > M5:
> >
> > ||$\tau=1$|2|3|4|5|6|Avg|Gain(%)|
> > |-|-|-|-|-|-|-|-|-|
> > |SCOT |6.4054+-0.0547|6.9328+-0.0634|7.2428+-0.0700|7.4585+-0.0580|7.7012+-0.0627|7.8278+-0.0651|7.2614+-0.0609|(-)|
> > |Temporal blocks only|6.4085+-0.0538|6.9547+-0.0535|7.2673+-0.0453|7.4825+-0.0388|7.7167+-0.0380|7.8328+-0.0430|7.2771+-0.0409|0.2%|
> > |Feature-wise blocks only|6.8805+-0.0333|7.6298+-0.0212|7.9706+-0.0254|8.1215+-0.0298|8.3989+-0.0411|8.5303+-0.0435|7.9219+-0.0311|8.3%|
> >
> > 8. **“Question 2: missingness in the time series.”**
> > In the Semi-Synthetic MIMIC-III dataset, we use the extracted features from [5]. [5] uses the so-called “Simple Imputation” described in its ”Data Pre-processing” section: missing values are first forward filled and then set to individual-specific mean if there are no previous values. If the variable is always missing for a patient, we set it to the global mean. In general, datasets can always be imputed to eliminate missing values. We also added the description of processing missing values in Appendix E.
> >
> > 9. **“Question 3: the model performance of pretraining without finetuning.”**
> > We first want to clarify that the supervised training stage on source domain data after pretraining is necessary for the outcome estimation task - the decoder outputting outcome estimation requires the supervised training stage. Our analysis of the generalization of our model to counterfactual estimation is also based on such a self-supervised pretraining -> supervised training scheme.
> >
> > After this step, we have three different settings depending on the domain of test data as well as whether we further finetune the model. (1) Non-Cold-Start (NCS): directly evaluate the model on source domain data; (2) Cold-Start (CS): directly evaluate the model on target domain data; (3) Transfer: finetune the model with a few samples from the target domain and evaluate on target domain data.
> >
> > We already report the performance in NCS, CS, and Transfer settings in Table 6 (in Appendix G), Table 1, and Table 2 respectively. While we are not fully certain what results the “pretraining without finetuning” is referring to, we assume Table 1 fits the description to some extent: the cold-start setting has no fine-tuning stage on target domain data. We hope the reviewer can clarify the question further.
> >
> > 10. **“Question 4: in Table 3 fold 2, the best one for MIMIC is actually sq. inv.”**
> > We thank the reviewer for pointing out the mark issue and we have fixed it in the revised version.
> >
> >
> >
> >
> > **References**:
> >
> > [1]. Melnychuk, Valentyn, Dennis Frauen, and Stefan Feuerriegel. "Causal transformer for estimating counterfactual outcomes." International Conference on Machine Learning. PMLR, 2022.
> >
> > [2]. Rehkopf, D. H., Glymour, M. M., & Osypuk, T. L. (2016). The consistency assumption for causal inference in social epidemiology: when a rose is not a rose. Current epidemiology reports, 3(1), 63-71.
> >
> > [3]. Bica, I., Alaa, A. M., Jordon, J., & van der Schaar, M. (2020). Estimating counterfactual treatment outcomes over time through adversarially balanced representations. In International Conference on Learning Representations.
> >
> > [4]. Robins, J., & Hernan, M. (2008). Estimation of the causal effects of time-varying exposures. Chapman & Hall/CRC Handbooks of Modern Statistical Methods, 553-599.
> >
> > [5]. Wang, S., McDermott, M. B., Chauhan, G., Ghassemi, M., Hughes, M. C., & Naumann, T. (2020, April). Mimic-extract: A data extraction, preprocessing, and representation pipeline for mimic-iii. In Proceedings of the ACM conference on health, inference, and learning (pp. 222-235).

---

> ### Author Response · Authors · 2023-11-22
> **Thanks for your thoughtful review! Looking forward to a fruitful discussion.**
>
> Thank you for your thoughtful and helpful review and suggestions! We have addressed your concerns and questions in our replies based on the review. We are looking forward to having an in-depth and fruitful discussion so that we can clarify any further confusion or concerns.

---

### Meta-Review · Area_Chair_AWQy · 2023-12-12

**Metareview:**

The paper presents an approach for using self-supervised learning for counterfactual estimation with a transformer. The goal is to improve representations of historical data. While the reviewers find the direction quite appealing, there were several issues raised in the discussion.

1. An unwillingness to share code raised concerns about reproducibility.

2. Over focus on the contribution being a transformer architecture. The main parts of the architecture exists in previous works, so this should not be considered the primary contribution.

3. Related to the second point was in framing. The reviewers were excited by self-supervised learning and thought that focus on self-supervised learning was an open area for which to make a clear contribution rather than the architecture framing that currently exists in the paper.

**Justification For Why Not Higher Score:**

The contribution of the paper is not clear and not sharing code raised questions about reproducibility.

**Justification For Why Not Lower Score:**

N/A

---

### Decision · Program_Chairs · 2024-01-16

Reject